# Challenging the Astral mass analyzer to quantify up to 5,300 proteins per single cell at unseen accuracy to uncover cellular heterogeneity

Julia A. Bubis [1,6] ✉, Tabiwang N. Arrey[2], Eugen Damoc[2], Bernard Delanghe[2], Jana Slovakova[3], Theresa M. Sommer[3,4], Harunobu Kagawa [3], Peter Pichler[1], Nicolas Rivron [3], Karl Mechtler [1,3,5] ✉ & Manuel Matzinger [1,6] ✉

Despite significant advancements in sample preparation, instrumentation and data analysis, single-cell proteomics is currently limited by proteomic depth and quantitative performance. Here we demonstrate highly improved depth of proteome coverage as well as accuracy and precision for quantification of ultra-low input amounts. Using a tailored library, we identify up to 7,400 protein groups from as little as 250 pg of HeLa cell peptides at a throughput of 50 samples per day. Using a two-proteome mix, we check for optimal parameters of quantification and show that fold change differences of 2 can still be successfully determined at single-cell-level inputs. Eventually, we apply our workflow to A549 cells, yielding a proteome coverage ranging from 1,801 to a maximum of >5,300 protein groups from a single cell depending on cell size and search strategy used, which allows for the study of dependencies between cell size and cell cycle phase. Additionally, our workflow enables us to distinguish between in vitro analogs of two human blastocyst lineages: naive human pluripotent stem cells (epiblast) and trophectoderm-like cells. Our data harmoniously align with transcriptomic data, indicating that single-cell proteomics possesses the capability to identify biologically relevant differences within the blastocyst.

Single-cell proteomics (SCP) by mass spectrometry (MS) has evolved into a powerful technique to investigate cellular heterogeneity with increasing coverage and throughput. To tackle the challenge of ultra-low input amounts, a multitude of miniaturized and automated sample preparation workflows have been developed by several labs[1–9]. The success of SCP, however, is also tightly connected to technological improvements in high-performance nanoflow liquid chromatography (LC) and high-resolution, high-sensitivity MS. The SCP field initiated its

success by using isobaric labels for multiplexing to improve sensitivity and throughput[1]. With more sensitive and fast mass spectrometers being commercialized, more and more groups focus on label-free quantitation workflows[10] and use data-independent acquisition (DIA) combined with short LC gradients. With these DIA workflows, optimal reproducibility across runs is achieved while maintaining throughput at acceptable levels ranging from 40 to up to 180 samples per day (SPD)[11–13]. Improvements in data analysis algorithms allow for confident

[1]Research Institute of Molecular Pathology (IMP), Vienna BioCenter, Vienna, Austria. [2]Thermo Fisher Scientific, Bremen, Germany. [3]Institute of Molecular Biotechnology (IMBA), Austrian Academy of Sciences, Vienna BioCenter, Vienna, Austria. [4]Vienna BioCenter PhD Program, Doctoral School of the University of Vienna and Medical University of Vienna, Vienna, Austria. [5]Gregor Mendel Institute of Molecular Plant Biology (GMI), Austrian Academy of Sciences, Vienna BioCenter, Vienna, Austria. [6]These authors contributed equally: Julia A. Bubis, Manuel Matzinger. ✉e-mail: julia.bubis@imp.ac.at; karl.mechtler@imp.ac.at; manuel.matzinger@imp.ac.at

identification of very-low-abundance peptides hidden within highly chimeric spectra.

The recent introduction of the Thermo Scientific Orbitrap Astral mass spectrometer[14] further extends the reachable sensitivity and acquisition speed by combining the established Thermo Scientific Orbitrap analyzer with the newly developed Thermo Scientific Astral analyzer in a single instrument. The Astral analyzer combines high speed (up to 200 Hz) at high resolution and sensitivity, with nearly loss-less transmission and a high dynamic range[14]. The instrument allows for high acquisition speed by simultaneously using the Orbitrap analyzer for $MS^1$ spectra and the Astral analyzer for $MS^2$ spectra at a resolution sufficient to resolve tandem mass tag reporter ions and at a sensitivity superior to the Orbitrap analyzer[14].

Here, we investigate the performance of the Orbitrap Astral mass spectrometer in combination with the Aurora Ultimate TS 25-cm nano-flow ultra-high-performance LC (UHPLC) column that comes with a fully integrated source interface to minimize peak broadening and yield the highest possible signal intensity. We optimize the gradient length to yield maximum proteome coverage and assess the accuracy and precision of quantification.

We apply our improved workflow to investigate the cellular heterogeneity of A549 human lung cancer cells. Additionally, we used optimized SCP on naive human pluripotent stem (hPS) cells, which recapitulate the preimplantation epiblast (EPI), and trophectoderm (TE)-like cells differentiated from naive hPS cells. During the in vitro fertilization process, only about 40% of fertilized eggs are estimated to reach the blastocyst stage with sufficient quality for transfer to the mother's uterus[15]. The criterion for selecting blastocysts largely depends on morphological characteristics, such as the expansion of the outer TE and a well-clustered inner EPI population[16,17]. However, our understanding of the mechanisms regulating blastocyst morphology remains limited. This limitation is primarily due to restricted access to human embryos and the lack of technologies capable of low-input detection methods. In this context, high-sensitivity SCP emerges as a powerful tool. It holds promise for unraveling the mechanisms underlying the regulation and functionalization of blastocyst morphology, including aspects like TE epithelialization, cavity formation and tissue segregation between the EPI and TE.

## Results

### Reaching unprecedented proteomic coverage

We combined the Orbitrap Astral mass spectrometer with the Thermo Scientific field asymmetric waveform ion mobility spectrometry (FAIMS) Pro interface and the Auroa Ultimate 25-cm TS column with an integrated emitter tip and zero dead volume, aiming to maximize sensitivity for ultra-low-input proteomic samples. To have a representative standard of comparable input to a single cell for evaluation and benchmarking, we decided to inject 250 pg of commercial HeLa and K562 cell peptides. The bulk digests were diluted in 0.1% trifluoroacetic acid (TFA) containing 0.015% *N*-dodecyl-β-D-maltoside (DDM) to improve peptide solubility, reproducibility and quantitation accuracy[18]. Gradient lengths were varied from 14 to 38 min (Supplementary Table 1), which corresponds to reachable throughputs of 30–80 SPD, including sample loading as well as column equilibration and washing time. During loading and washing, flow rates were increased to improve throughput, and lower flow rates were applied during peptide elution to enhance sensitivity[13,19,20]. As shown in Fig. 1a,e, we found a sweet spot around 50 SPD, yielding maximal identifications. Of note, the data were analyzed library free. When replicates were searched together, the number of protein groups (PGs) identified was boosted by 6.3–10.9% and 4.5–7.2% for HeLa cell peptides and K562 cell peptides, respectively. In addition, by allowing for matching, replicate measurements nicely align, leading to a data completeness close to 100% and vanishing error bars for ID numbers compared to the method evaluation mode.

Based on our results, we hypothesize that shorter gradients are advantageous for ultra-low-input samples as they produce sharper, and hence more intense, chromatographic elution peaks. Once a maximum is reached at 50 SPD, faster gradients potentially suffer from insufficient separation power and greater spectrum complexity, which adversely affects identification numbers. Fast gradients with very narrow elution peaks further reduce the number of data points per peak (Fig. 1c,g). We decided to focus on 50 SPD as the gradient length for all further experiments within this study as it still shows the best proteome coverage with five data points per peak on median ($MS^1$ level), enough for proper integration. We further decided to quantify on the $MS^1$ level as we see not only more data points there but also a lower median coefficient of variation (CV) clearly below 10% (Fig. 1b,f).

Next, we created a tailored library from 10 ng of the very same HeLa and K562 digests to further improve proteome coverage. The libraries were also recorded in DIA mode and resulted in more than 62,000 precursors identified within the library (Fig. 1d,h). Using these libraries, we performed a library search for the 250-pg runs, which improved PG identifications by 41.5% and 31.7% compared to a library-free search in method evaluation mode and by 26.3% and 25.6% compared to the library-free search with matching for HeLa and K562 cell peptides, respectively (Fig. 1d,h). Using default settings for the Spectronaut search engine, we identified more than 7,500 proteins from a single run with as little as 250 pg of HeLa cell peptide input and more than 6,800 proteins with 250 pg of K562 cell peptides. These numbers increased further to a total of 7,800 and 7,060 unique PGs identified from three replicates of HeLa and K562 cell peptides, respectively (Supplementary Fig. 1a,b). Of note, we identified a total number of 6,126 PGs from HeLa cell peptides using DirectDIA+ in method evaluation mode and only 6,017 using DirectDIA+ with matching replicates (DirectDIA+). We hypothesize that this is reasoned by a global false discovery rate (FDR) control applied over all three replicates on top of the run-specific control while only very few additional proteins could be matched in those technical replicates. Of those 6,017 total proteins, 5,989 were found in all replicates, indicating excellent data completeness (Supplementary Fig. 1a). We additionally checked for potential FDR inflation by using a library to boost ID numbers and performed a cosearch of 250-pg and 10-ng files instead (Supplementary Fig. 1c). When doing so, the PG numbers were similar to each other (difference of -0.4%), and at the precursor level, the difference was -5%, suggesting that FDR inflation is no problem in the Spectronaut version used (18.6).

Importantly, our results depicted a single hit wonder and were also successfully reproduced on a second Orbitrap Astral run using a different batch for the analytical column and K562 digest (Supplementary Fig. 1d).

To exclude potential issues occurring from carryover when using shorter gradients, we assessed the degree of residual protein and precursor identifications in the first five washes after 250-pg injections. (Supplementary Fig. 2). In method evaluation mode, we were not able to identify any proteins in wash runs where only 0.1% TFA was injected, as too little spectra for calibration were available. When allowing for matching across the recorded replicates, the first wash run after a sample injection yielded <2% of precursor IDs compared to the previous 250-pg injection. The total quantity of protein in those first washes was more than two orders of magnitude lower than the previous 250-pg injections independent of the gradient length used. Although those quantities are further reduced following continued washing, we conclude that the level of carryover is neglectable for such single-cell-level samples and for the throughput range tested in this study. Furthermore, we did not see any accumulation of material on the column over time (that is, no increase in quantity from the first to fifth replicate of a 250-pg injection; Supplementary Fig. 2c).

We assume that 250 pg of HeLa cell peptides is equivalent to single-cell-level input for methodological comparisons in this study; however, protein contents ranging from 100 to 500 pg were reported

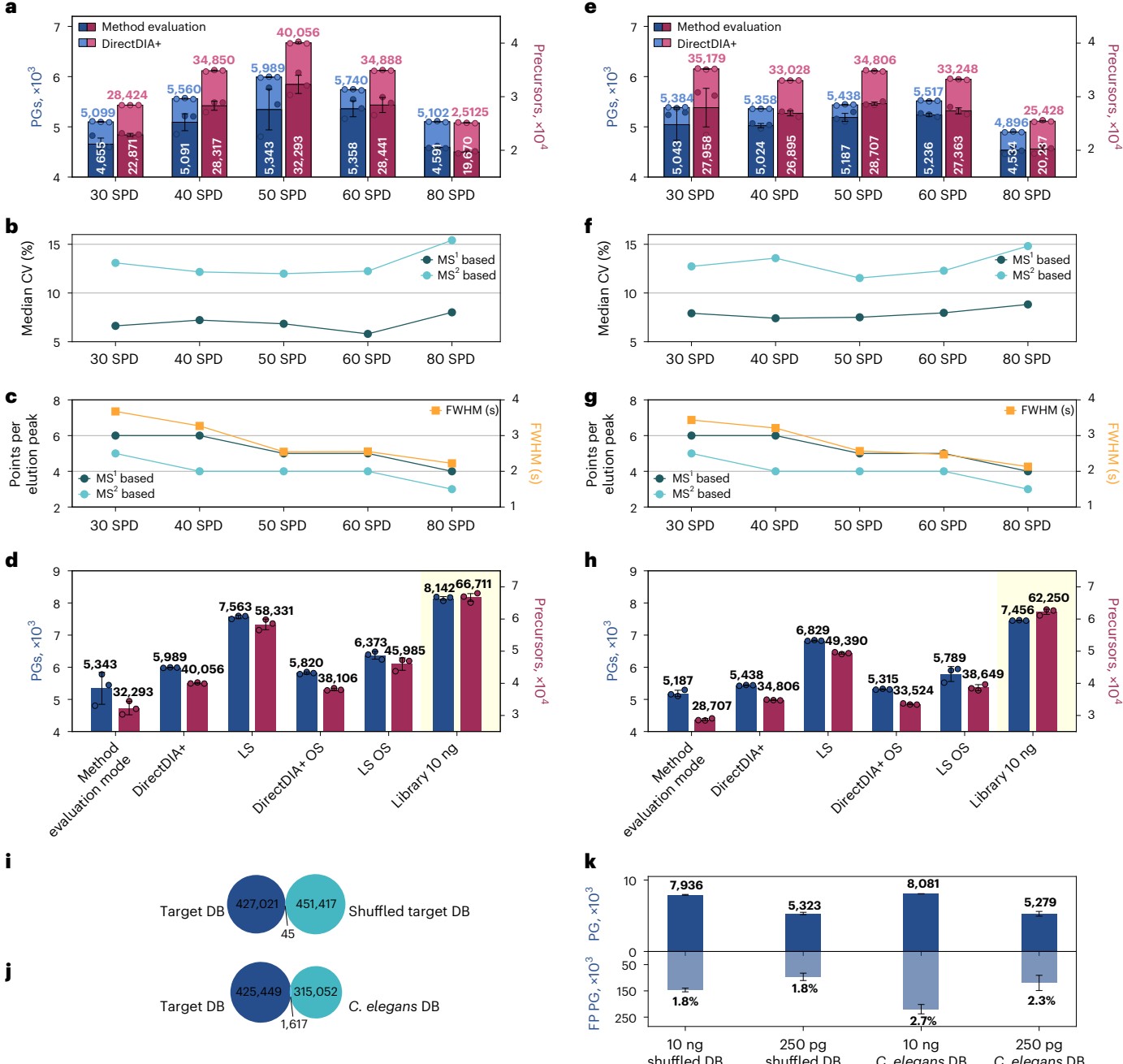

**Fig. 1 | Finding optimal parameters to maximize PG IDs and maintain data quality. a–h,** Two hundred and fifty picograms of HeLa (**a–d**) or K562 (**e–h**) cell peptides from diluted bulk digests were injected. Peptides were separated using a gradient of altered length ranging from throughputs of 30 to 80 SPD. Data were recorded in DIA mode on an Orbitrap Astral mass spectrometer. Circles in **a** and **e** indicate identified PGs or precursors at a 1% FDR in individual replicates, bars indicate their means, and error bars indicate standard deviation with $n$ = 3 technical replicates. Dots in **b** and **f** indicate the obtained median CV on protein levels for each gradient length for DirectDIA+ analysis when performing quantification at the MS[1] or MS[2] level, respectively. Dots in **c** and **g** indicate the median number of data points per precursor that were used for quantification (DirectDIA+) for each gradient length at the MS[1] or MS[2] level, respectively, as well

as the median full-width at half-maximum (FWHM) of elution peaks. Different search strategies on 250 pg of HeLa (**d**) and K562 (**h**) 50 SPD data at a 1% FDR and default settings or optimized settings (OS) were compared. The last bar 'Library 10 ng' corresponds to the tailored library itself that we used for a library search (LS) of 250-pg runs. **i,j,** For FDR assessment, data were analyzed in method evaluation mode at a 1% FDR using default settings against a target (that is, the human proteome), a shuffled target (that is, a shuffled human proteome; **i**) or a *C. elegans* database (**j**). Venn plots show the overlap of peptides between target and shuffled or entrapment databases, respectively. **k,** Bars indicate means of identified PGs in the target database (top) and false-positive PGs (FP PG) from the shuffled target or *C. elegans* database (bottom). Error bars indicate standard deviation with $n$ = 3 technical replicates. DB, database.

for HeLa cells depending on their size and cultivation time[21]. We therefore checked reachable proteome coverage for a broader range of input amounts ranging from 50 pg to 10 ng (relevant when using libraries or carrier proteomes). This resulted in ~4,700 to ~8,300 PGs identified

from 50 pg to 10 ng, respectively (Supplementary Fig. 3). To consider the large difference in sample input, we adopted an isolation window and maximal injection time ranging from 5 to 20 Th and 10 to 80 ms, respectively. We consider such an adoption as highly relevant also for

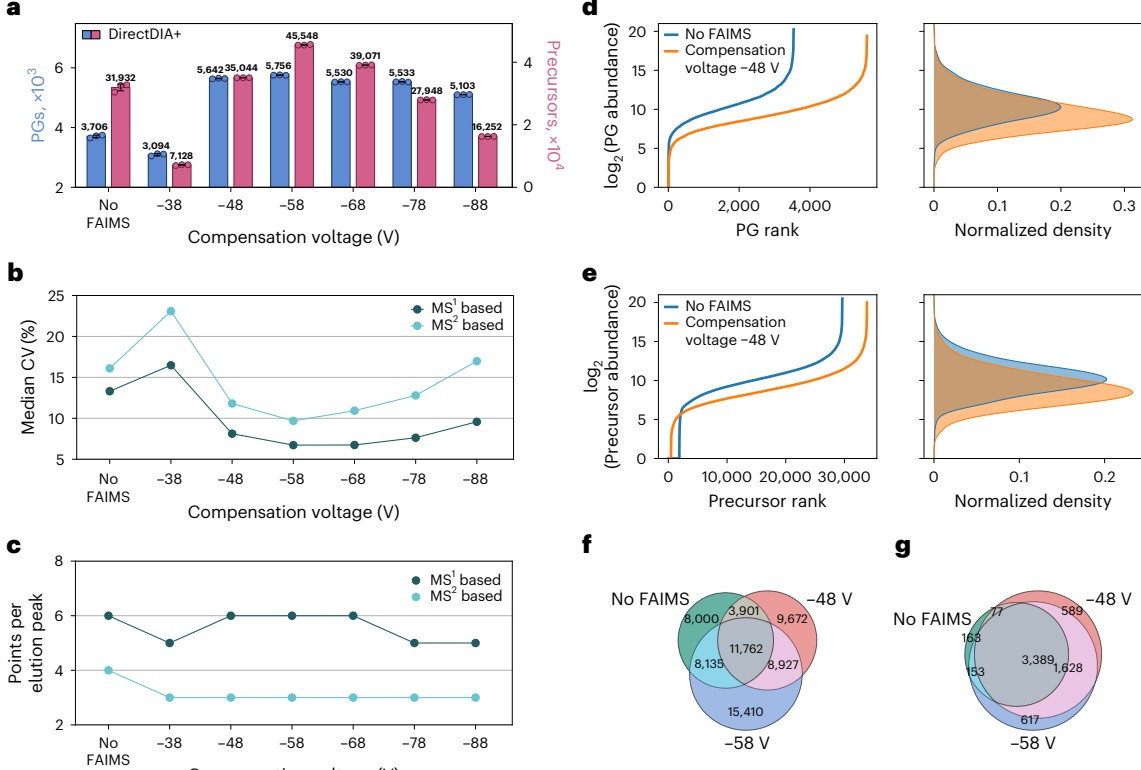

**Fig. 2 | Tuning coverage and signal to noise by FAIMS. a–g,** Two hundred and fifty picograms of HeLa cell peptides from diluted bulk digests were injected. Peptides were separated at a throughput of 50 SPD; $n = 3$ technical replicates. Data were recorded with or without the FAIMS Pro interface unit attached and at the given compensation voltage. Circles in **a** indicate identified PGs or precursors at a 1% FDR in individual replicates, bars indicate their means, and error bars indicate standard deviations. Dots in **b** indicate the median CV at the protein level when performing quantification at the MS[1] or MS[2] level using Spectronaut 18 (DirectDIA+). Dots in **c** indicate the median number of data points recorded per precursor when performing at the MS[1] or MS[2] level as indicated (DirectDIA+). The $\log_2$ abundance ranks and their density plots for PGs (**d**) and precursors (**e**) are shown. The Venn diagrams show the number of PGs (**f**) or peptides (**g**) quantified in one representative replicate each using either no FAIMS or FAIMS with a compensation voltage of −48 V or −58 V.

the generation of tailored libraries from higher input (see Fig. 4 without an adopted acquisition method and Fig. 5 with an adopted acquisition method for real single cells).

**Validation of the FDR strategy**

To estimate the real FDR, we generated a shuffled target database by shuffling all sequences of our target database while maintaining the positions of all protease cleavage sites (P, K and R). Both databases were used as targets in the subsequent search. They have less than 0.01% shared peptides and a similar size, allowing for a fair FDR estimation with a very low risk of finding peptides in the shuffled target database that represent true-positive hits (Fig. 1i). Of note, randomization of peptide sequences led to a slightly increased number of unique peptides in the shuffled target versus the target database, which we opted for deliberately to ensure a conservative estimate of the FDR.

When examining the HeLa cell data presented in Fig. 1d, we found that the FDR at the PG level was slightly above the expected 1%, indicating that default settings in Spectronaut version 18 are too relaxed (Fig. 1k). This is in line with earlier results[22] demonstrated in Spectronaut versions 16 and 17, where the FDR was reported to be higher than expected when using default parameters.

We additionally used a *Caenorhabditis elegans* database as a second entrapment strategy for FDR validation (Fig. 1j). The estimated FDR when using *C. elegans* as entrapment was a bit higher and above 2%, confirming our initial observation of potentially too relaxed settings. We therefore tested the application of a more stringent cutoff of 0.01 for the run-wise protein $q$ value, which did, however, not alter the results. Instead we applied fully stringent settings suggested by

Baker et al.[22]. Their optimized settings set all cutoff values to 0.01. As shown in Fig. 1d,h, the additional protein $q$ value cutoff did not change the results in DirectDIA mode, and numbers dropped only slightly for the library search. Using the settings suggested by Baker et al., PG numbers for HeLa cells dropped by ~3% for runs in DirectDIA+ and by ~16% for library searches.

Of note, reporting total ID numbers by summing up all unique IDs from three replicates (Supplementary Fig. 1a,b), without an additional FDR filter step, further accumulates wrongly identified precursors and proteins. This leads to an elevated FDR of up to 5.5% (Supplementary Fig. 1e). Hence, we advise reporting average ID numbers rather than total ID numbers.

**The FAIMS Pro interface improves signal to noise, CV and coverage**

Aiming for the best possible sensitivity, we screened for the optimal compensation voltage in high FAIMS using the FAIMS Pro interface. From previous experience, −48 V yields optimal results on the used device, which is why that voltage was used for all other runs. Screening the compensation voltage from −38 to −88 V in steps of 10 suggests that the optimal setting is between −48 and −68 V, with few more PGs and clearly more precursors detected at −58 V than at any other setting. Although using the FAIMS Pro interface bears the risk of losing some ions, it still seems advantageous due to an improved signal-to-noise ratio and dynamic range[23,24]. As reported earlier by our lab[25], a clear gain in sensitivity, especially for limited-sample LC–MS/MS analyses, can be seen by using the FAIMS Pro interface, as the relative contribution of singly charged background ions gets more substantial at low sample loads.

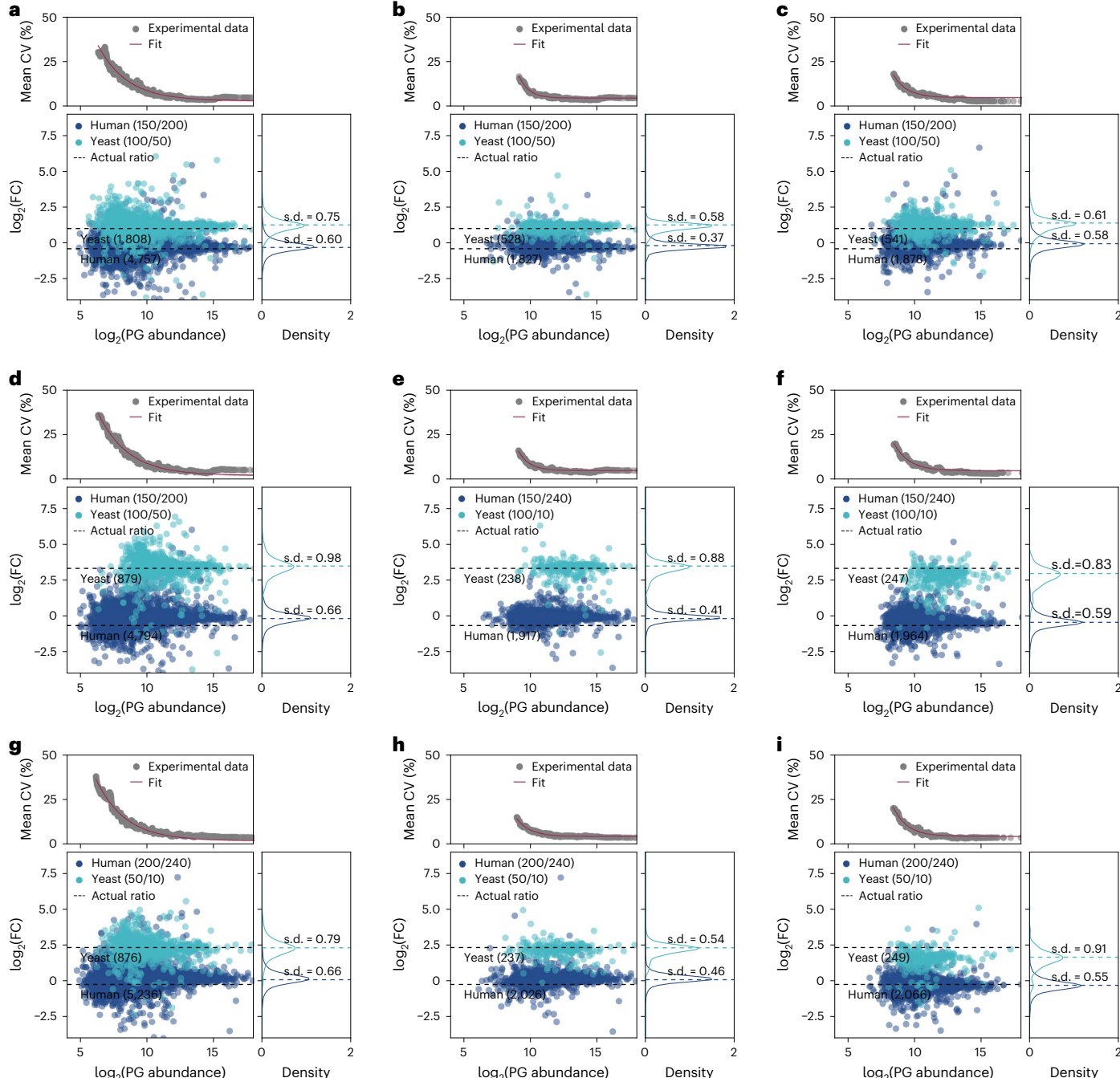

**Fig. 3 | Human–yeast proteome mix to assess quantitative performance.**
**a–i**, From diluted bulk digests, 250 pg of two-proteome mixes consisting of 150 pg of HeLa + 100 pg of yeast, 200 pg of HeLa + 50 pg of yeast and 240 pg of HeLa + 10 pg of yeast were injected. Peptides were separated at a throughput of 50 SPD. Data were recorded in DIA mode using optimal, but not the same, settings for the Orbitrap Astral mass spectrometer and Orbitrap Exploris 480 mass spectrometer and analyzed using DirectDIA+ in Spectronaut 18 at a 1% FDR. Quantification was performed at the MS[1] level. Dots within the Bland–Altman plots (bottom) represent proteins with given $\log_2$ average PG abundance and

$\log_2$ fold change of abundance across both proteome mixes. Density plots (right) depict the distribution of measured $\log_2$ fold change (FC) values, and CV diagrams (top) show the local CV of 100 proteins quantified with a rolling window over the entire abundance range; $n$ = 3 technical replicates. For the Orbitrap Astral mass spectrometer, all quantified proteins (**a**, **d** and **g**) or only those proteins that were commonly quantified using the Orbitrap Exploris 480 mass spectrometer (**b**, **e** and **h**) are shown. For the Orbitrap Exploris 480 mass spectrometer, all quantified proteins are shown (**c**, **f** and **i**).

In our hands, use of the FAIMS Pro interface at optimal settings resulted in a clearly improved signal-to-noise ratio (Supplementary Fig. 4), with up to 42.6% more precursors and 55.3% more PGs identified than measurements without a FAIMS interface (Fig. 2a). In line with our expectation from the improved signal-to-noise ratio, we quantified more low-abundance precursors/proteins when using FAIMS

(Fig. 2d,e). However, some of the abundant precursors were lost, resulting in only 49% and 63% of peptides identified in runs without FAIMS being found at the most promising compensation voltages of −48 V and −58 V, respectively (Fig. 2f). Among shared peptides between runs without the FAIMS Pro interface and with a compensation voltage of −48 V, fewer close to 0 or missed values were present in the

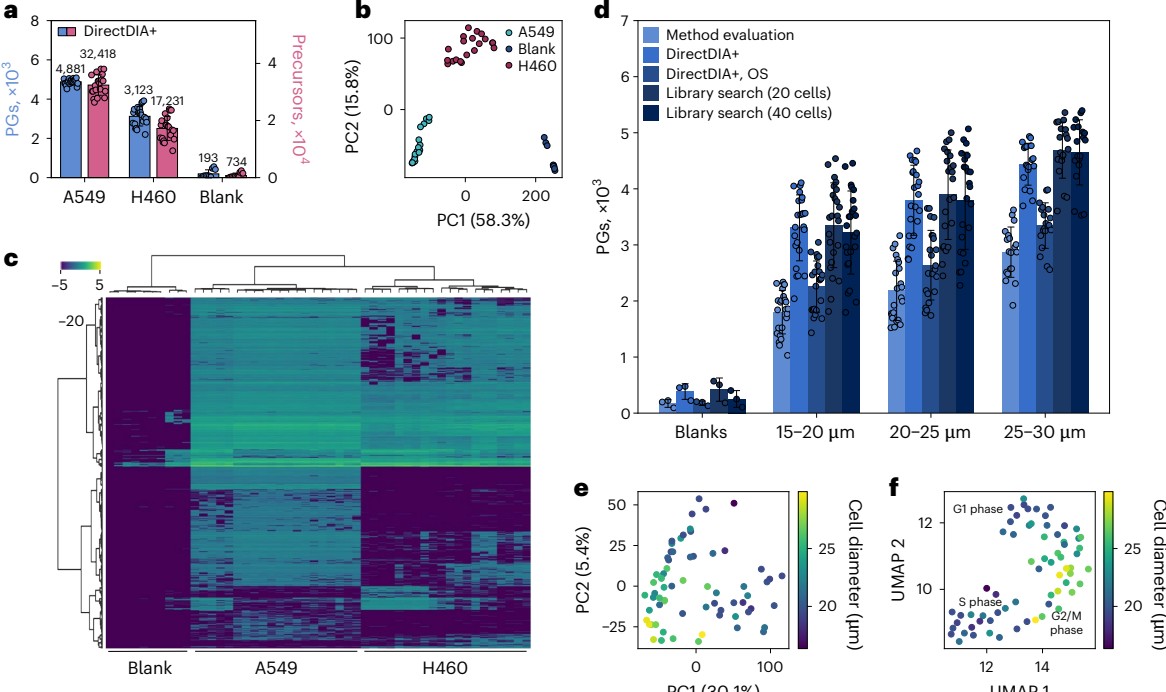

**Fig. 4 | Heterogeneity of single A549 and H460 epithelial-like human lung cancer cells. a–f,** Individual cells of 20–30 μm in diameter were prepared using the One-Pot workflow and analyzed at 50 SPD using the previously optimized settings for LC and MS on the Orbitrap Astral mass spectrometer. First, A549 and H460 cells were analyzed. In **a**, circles indicate identified PGs or precursors at a 1% FDR of each individual cell, bars indicate mean values, and error bars indicate standard deviations. The search was performed library free with matching across biological replicates but not across all samples; *n* = 20 individual cells for each cell type. The PCA (**b**) was based on protein quantities, with each dot representing a cell or blank sample, and heat map clustering (**c**) was performed with each column representing a cell or blank sample and color codes depicting relative protein

abundance. In total, *n* = 66 A549 cells in predefined size groups of 15–20 μm, 20–25 μm or 25–30 μm in diameter were analyzed (**d–f**). In **d**, circles indicate identified PGs at a 1% FDR of each individual cell, bars indicate the mean values, and error bars indicate standard deviations. The search was performed library free in method evaluation mode, DirectDIA+ or DirectDIA+ optimized settings or against a tailored library created from three replicates of 20 or 40 cells. The PCA (**e**) and UMAP (**f**) are based on protein quantities (DirectDIA+ analysis) of the 66 A549 cells, with each dot representing a cell and colors reflecting the actual cell size as determined by the camera system of the cellenONE. All samples (**a–f**) were prepared and measured in the exact same way. Blanks were processed in the same 384-well plate and contained all reagents and buffers but no cells.

data recorded with the FAIMS interface (Supplementary Fig. 5b,c). Almost all (98%) peptides found in runs without FAIMS were present in one of the compensation voltage runs (Supplementary Fig. 5a). The observed effect was less dramatic at the protein level, where almost all proteins quantified without FAIMS were also quantified with FAIMS (Fig. 2g). In addition, FAIMS usage yielded more identified proteins. Global sequence coverage with FAIMS was slightly reduced (that is, on average, 7.8 peptides/protein were identified without FAIMS but only 7.6 and 6.0 peptides/protein at compensation voltages of −48 V and −58 V, respectively). Acquisition speed was also not affected as there was no influence on the number of data points per peak (Fig. 2c), and the CV (Fig. 2b) was strongly improved by FAIMS. Overall, we believe that FAIMS usage is highly advantageous for most single-cell studies mainly due to the improvement of signal-to-noise and CV that results in more proteins being identifiable and quantifiable.

## Quantitative performance

We next evaluated the quantitative performance of our workflow at single-cell-level inputs and benchmarked the data to our Thermo Scientific Orbitrap Exploris 480 mass spectrometer using similar settings, the same input and the same gradient length.

Although the CV gives a reasonable estimate of the quantitative precision across technical replicates, we further evaluated the accuracy and precision of proteins quantified from a two-proteome mix (Fig. 3). By comparing two proteome mixes with different, but known, human-to-yeast mixing ratios, we assessed the accuracy of quantification. Of note, less than 50 pg of yeast, present in one of the samples,

still enabled the quantification of 1,782 yeast proteins. As shown in Fig. 3a, the Orbitrap Astral mass spectrometer does an excellent job delivering a fold change in protein abundance very close to the expected value and outperforms the Orbitrap Exploris 480 mass spectrometer, especially for the more sparse yeast samples (Fig. 3). We further checked accuracy by looking at the distribution around the expected fold change and found that higher-abundance proteins within our sample tended to be quantified very accurately with a local CV of less than 5%. The local CV, however, rose with lowered protein abundance and approached 35% at the limit of quantification. We observed an exponential drop of CV with increasing protein abundance.

Our results show that on the Orbitrap Astral mass spectrometer, more than twice the number of proteins was quantified compared to on the Orbitrap Exploris 480 mass spectrometer, which is likely due to its improved speed and sensitivity. When filtering for commonly found proteins (Fig. 3b,h,i), predominantly lower-abundance proteins were missing, highlighting the increased sensitivity of the Orbitrap Astral mass spectrometer. Furthermore, almost all proteins identified with the Orbitrap Exploris 480 mass spectrometer were also identified using the Orbitrap Astral mass spectrometer. The number of common proteins was very close to the protein number quantified using the Orbitrap Exploris 480 mass spectrometer. The commonly quantified proteins allow for a fair comparison of accuracy and precision between the Orbitrap Astral mass spectrometer and the Orbitrap Exploris 480 mass spectrometer and shows that the distribution of fold change values is clearly smaller on the Orbitrap Astral mass spectrometer. The same is true for the local CV and is most distinct for the proteins

of lowest abundance, where the CV drops from ~20% for the Orbitrap Exploris 480 mass spectrometer to ~12% for the Orbitrap Astral mass spectrometer. Because quantification was performed at the $MS^1$ level, data from both instruments originated from an Orbitrap analyzer of the same construction type. We hypothesized that the increased CV even for the very same proteins is due to the lower speed of the Orbitrap Exploris 480 mass spectrometer. The Orbitrap Exploris mass spectrometer uses the Orbitrap analyzer for both $MS^1$ and $MS^2$ data acquisition, whereas the Orbitrap Astral mass spectrometer simultaneously uses the Orbitrap analyzer for $MS^1$ and the Astral analyzer for $MS^2$, which gives the latter an enormous advantage in speed. This is also reflected in more data points per elution peak recorded on the Orbitrap Astral mass spectrometer (median of three for Exploris and five on the Astral for this dataset).

Applying an additional filter to exclude proteins quantified based on only one peptide had no impact on the accuracy but further improved the quantitative precision, independent of the instrument used (Supplementary Fig. 6). As a tradeoff, 18% (Orbitrap Astral mass spectrometer) to 7% (Orbitrap Exploris 480 mass spectrometer) fewer proteins were quantified for a 2:1 ratio mixture. We investigated the quantitative performance using larger fold changes of 5:1 and 10:1 in yeast quantity (Fig. 3d–i), which showed similar results.

We additionally checked quantitation performance at the $MS^2$ level, yielding a bit wider deviation around the expected fold change ratio than seen at the $MS^1$ level (Supplementary Fig. 7).

## Proof-of-principle studies using single cells

**Heterogeneity of two human non-small cell lung cancer models.**
We first benchmarked several previously reported sample preparation workflows to decide on an optimal strategy for our single-cell studies (Supplementary Information and Supplementary Fig. 8). We decided on the One-Pot[7] workflow as it delivered optimal coverage and low background levels. We applied our optimal workflow conditions for sample preparation and data acquisition to single cells from lung cancer. The A549 and H460 cell lines are epithelial-like human non-small cell lung cancer cell lines commonly used for basic research and drug discovery. Although the same size range (20–30 μm) of cells was selected for both cell types, their protein content seemed surprisingly different. Although we yielded 4,879 average PGs and 32,417 precursors for A549 cells, coverage for H460 cells was clearly lower at 3,166 average PGs and 17,619 precursors (Fig. 4a). In line with our previous results (Supplementary Fig. 8), blank controls showed only low background signals besides the abundant trypsin peaks. Linear dimensionality reduction through principal component analysis (PCA) and heat map analysis showed a clear separation of both cell lines and from the blank control runs based on their protein abundance profiles, proving the capability of the method to investigate cellular heterogeneity (Fig. 4b,c).

**Size: cell cycle dependency of A549 cells.** To investigate our A549 model in more detail, three different size groups of cells ranging from 15–30 μm in diameter were collected using the cellenONE robot. Sixty-six single A549 cells and three blanks were measured using the Orbitrap Astral mass spectrometer. As expected, the number of PGs identified correlated with the size of the cells (Fig. 4d) and ranged from 1,801 PGs for small cells to 2,870 PGs for large cells, on average, when searching library free and in method evaluation mode. When allowing for matching across single cells (DirectDIA+), ID numbers were improved to 3,304 (+83%) and 4,439 (+55%) for smaller and larger cells, respectively. These average ID numbers are very close to numbers we obtained for A549 cells in our first single-cell dataset (Fig. 4a), which was measured on a different Astral instrument and in a different lab, demonstrating the reproducibility of the workflow. We additionally investigated if the use of a spectral library might further increase ID numbers and created a tailored library either from 20 or 40 A549 cells isolated in a single well of a 384-well plate, processed and measured

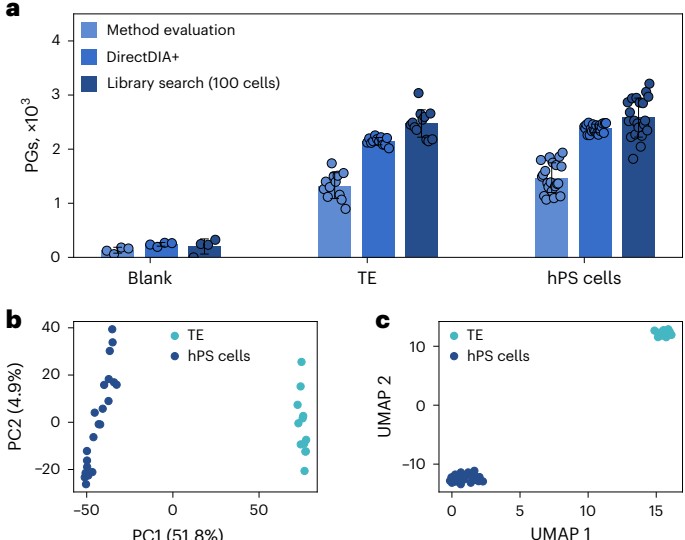

**Fig. 5 | Analysis of single TE and hPS cells.** Individual cells were isolated using FACS into a 384-well plate. Digested cells were analyzed at 50 SPD using the optimized settings for LC and MS on the Orbitrap Astral mass spectrometer. **a**, Circles indicate identified PGs at a 1% FDR of each individual cell, bars indicate the mean values, and error bars indicate standard deviations. The dataset contains 21 and 12 individual hPS cells and TE-like cells, respectively. The search was performed library free in method evaluation mode or DirectDIA+ or against a tailored library created from three replicates of 100 cells each prepared in the exact same way but recorded with adopted settings (that is, 60 ms IT and $m/z$ 20 windows for single cells, 10 ms IT and $m/z$ 5 windows for 100 cells). Blanks ($n = 4$) were processed in the same 384-well plate and contained all reagents and buffers but no cells. **b,c**, The PCA (**b**) and UMAP (**c**) are based on protein quantities, with each dot representing a cell and colors reflecting the cell type based on fluorescent marker proteins.

using the exact same settings as for single cells. We could not significantly improve proteome coverage using a library search instead of DirectDIA+, although the used 20- and 40-cell libraries contained much more protein than found in single cells (6,395 PGs for 20 cells and 6,762 PGs for 40 cells). Using the 40-cell library, ID numbers even dropped a bit (Fig. 4d). This is why we assume that there is a sweet spot in optimal library size. MS acquisition parameters, optimized for single-cell input, as accumulation time or $m/z$ windows are likely not perfectly optimized anymore for larger amounts as used for a large library. In addition, we believe that the high number of replicates already serves as an excellent basis for matching. In line with those assumptions, we observed a more pronounced effect of using a library when optimizing the MS method for the higher input and when using fewer replicates of single cells, as done in our second study on TE-like cells and naive hPS cells (Fig. 5). In our hands, using a 20-cell library yielded the best results and enabled us to identify, on average, 3,351 PGs from small single A549 cells and 4,689 PGs from large single A549 cells. Maximum coverage was 5,300 proteins from a single A549 cell. We also examined the performance of DirectDIA+ with optimized settings, which yielded slightly more ID numbers than the method evaluation approach.

We investigated the technical characteristics of the dataset, data completeness and CV between single cells at the PG level. Data completeness dropped dramatically, with the number of files reaching only 1,053 of 5,805 PGs (-18%) identified in all 66 files (Supplementary Fig. 9a). There was also a slight dependence on cell size; for the 15- to 20-μm cell diameter range, we observed slightly lower data completeness across 19 files than in the two other groups (20–25 μm and 25–30 μm; Supplementary Fig. 9b–d). As expected, mainly due to biological heterogeneity, the CV distributions for this dataset also looked completely different from HeLa or K562 cell peptide dilution

series, where the CV was below 10%. The median CV distribution for single-cell runs was 65%, whereas for stock runs of 20 and 40 cells the median CV distributions were 17% and 16%, respectively (Supplementary Fig. 9e–g). This difference in CV aligns with our assumption that cells have a unique protein abundance distribution, which results in a high CV among single cells. In 20- and 40-cell samples, the CV was much lower because we averaged among 20 or 40 cells; however, it was not close to HeLa or K562 cell peptide dilution series. Several factors might be the issue: (1) 20 or 40 cells is not enough to average all biological differences between cells, and (2) for real datasets, sample lysis and digestion cause some fluctuations.

We next examined the extent of cellular heterogeneity of these cultured and untreated A549 cells by PCA and found that cells clustered differently dependent on their diameter (Fig. 4e). This is even more pronounced when performing a nonlinear uniform manifold approximation and projection (UMAP; Fig. 4f). We took a deeper look at the most abundant proteins identified in the single cells, which were histone H4 (H4C1) and the actins ACTB and ACTC1. Indeed, their abundance profiles were nicely reflected in the clustering seen in the PCA and UMAP (Extended Data Fig. 1). We suspect that their abundance differences might correlate with not only the recorded cell sizes but also the different cell cycle stages. G2/M phase cells are reported to be the largest[26], which is why we annotated our largest cell size cluster as presumably G2/M. Next, we tried to confirm this by checking the expression pattern of abundant and known marker proteins. Actins are known to play a crucial role in cell division and in the formation of cell junctions and maintenance of cell shape, which is highly relevant especially for cancer cells and therefore why it seems logical that their expression levels correlate to cell size and cell cycle stage[27,28]. Histone levels are known to be elevated during G2/M phase when DNA synthesis occurs. To determine whether the cells with elevated H4C1 and H14 levels as determined by PCA and UMAP (Extended Data Fig. 1) could indeed be primarily in G2/M phase, we investigated the abundance levels of the nuclear ubiquitous casein and cyclin-dependent kinase substrate 1 (NUCKS1), which was also quantified in this dataset. NUCKS1 is known to be expressed at high levels in S phase[29,30] and is indeed present at lower expression levels in those cells presumably in G2/M phase but is upregulated in another cell cluster (presumably in S phase). We also assessed the expression of the classical cell cycle marker CDK1. CDK1 was reported to have similar protein expression levels as NUCKS1 (ref. 29), but we could not see a clear trend in the data. MKI67, another reporter protein expected to be maximally expressed in G2 phase[31], also showed results correlating to the size distribution (Extended Data Fig. 1).

Of note, no cell cycle control was performed in this study, which is why our cellular population reflects a wild mix of all phases, impeding a clear separation but still allowing for the successful study of the heterogeneity of this system.

**Characterization of human hPS cells and TE-like derivatives.** To challenge our analytical LC–MS/MS setup and the abilities of SCP further, we analyzed naive hPS cells and induced from them TE-like cells. On day 4, TE-like cells were collected and sorted for expression of the TROP2 marker protein using fluorescence-activated cell sorting (FACS). We evaluated different analysis strategies (Fig. 5a). With our library-based approach, we could identify 2,339 and 2,544 PGs in TE-like and naive hPS cells, respectively. Data completeness at the PG level was 53% for hPS cells and 60% for TE-like cells across all files (Supplementary Fig. 10a,b). The CV distribution of PG abundances for TE-like and hPS single cells had a median of 44%, whereas 100-cell runs had abundances of 13% and 13%, respectively (Supplementary Fig. 10c–f).

To confirm that SCP can recapitulate known differences between naive hPS cells and TE-like cells, we performed PCA and UMAP clustering analysis on all cells in the dataset. We observed strong separation between these two cell populations (Fig. 5b,c) with both approaches,

which aligned with cell type. PCA reported an explained variance of 51.8% (PC1) and 4.9% (PC2).

Gene Ontology (GO) analysis of proteins upregulated in TE-like cells showed enrichment in the following biological processes: cytoskeleton organization (GO:0007010), actin cytoskeleton organization (GO:0030036), epithelial cell differentiation (GO:0030855), epithelium development (GO:0060429), positive regulation of cell differentiation (GO:0045597) and cell differentiation (GO:0030154), which rightfully corresponds to the previously described biological processes underlying the differentiation of hPS cells into TE-like cells[32] (Supplementary Fig. 11).

TE cell analogs express the proteins GATA2, GATA3, CDX2 and TROP2 (ref. 33). Transcription factors are typically of low abundance in the cell and are often challenging to detect with proteomics techniques. They are also partly localized in the nucleus, which can make them less accessible for extraction and analysis. Additionally, many transcription factors are modified by phosphorylation or other post-translational modifications that can affect their detection. Here, we were able to identify the proteins GATA2 and GATA3 at very low abundance in some TE-like cells. Among differentially expressed proteins between hPS cells and TE-like cells, we found many TE markers well known from transcriptome data: KRT18 (epithelium cytoskeleton), KRT19 (epithelium cytoskeleton), RAB25 (RAS pathway), DAB2 (ERK pathway), YAP1 (Hippo pathway), S100A16, SP6, CDH1, ENPEP, SLC7A2, HAVCR1 and PDLIM1, which are associated with TE development, and DPPA4, DPPA2, SUSD2 and DNMT3L, which are well-established naive hPS cell markers[32,34] (Extended Data Fig. 2). Abundances for these proteins correlate with transcriptome data known from the literature[32,34].

Single-cell proteome data of naive hPS cells and TE-like cells demonstrate the capacity to distinguish primary cells and to extract meaningful biological information that complements and enriches results from transcriptome data.

**Comparison with pseudobulk data (100 cells).** We compared the single-cell dataset of TE-like and hPS cells to pseudobulk data generated from 100-cell stocks. PCA distinguished different cell types with explained variances of 54.1% (PC1) and 15.5% (PC2; Supplementary Fig. 12a). We examined protein abundances of marker proteins, which were found for single-cell data. For all proteins but one, the trend was the same. ENPEP was found to be abundantly expressed in both TE-like and hPS cells in bulk data; however, at the single-cell level, it was found at high abundance only in TE-like cells.

GO analysis of proteins upregulated in the TE-like bulk dataset showed much more biologically relevant pathways, even though the total number of upregulated proteins in pseudobulk data (from 100 cells) was lower than in single-cell data (415 versus 243 proteins). If we compare the same pathways that were identified in single-cell and bulk data (Supplementary Figs. 11 and 13), the number of proteins in enriched pathways was similar; however, the P values of the pathways for the pseudobulk data were lower. This is likely because of more intense, and hence more confident, signals and results in a much clearer picture originating from the bulk dataset than the noisier single-cell data. The intersection of upregulated proteins between bulk and single-cell data is 60 proteins.

## Discussion
Despite tremendous improvements already made in the field of ultra-low-input proteomics by MS in the past decade, achieving a high sample throughput, reproducibility, quantitative accuracy, precision and high sensitivity at the same time is still very challenging. The presented workflow aims to fulfill those criteria by combining a high-performance zero-dead-end volume column at a short gradient with one of the most advanced, fast and sensitive MS instruments, FAIMS-based noise reduction and a DIA-based acquisition. The used packed bead column has a pulled emitter directly at the end of the

column, which minimizes peak broadening and is further supported by the use of high flow rates during sample loading, increasing the throughput. LC flow rates are reduced during the active gradient to enhance sensitivity. We found that total throughputs of 50 SPD represent a sweet spot between maximizing speed of sample acquisition and maintaining enough measurement time to detect a median of five data points per elution peak, enabling proper quantification.

Here, we assess quantitative accuracy and precision at the single-cell level using a two-proteome mix. At five data points per elution peak as reached on the Orbitrap Astral mass spectrometer at 50 SPD, successful separation of twofold changes in protein abundance and quantification of as little material as 10 pg in the most extreme case still worked. However, the CV of quantified proteins within a sample increased with decreased protein abundance. Hence, relative abundance within a single-cell sample is an important consideration for future biological studies as the confident clustering of cellular populations can currently only rely on more abundant proteins within single cells. Despite this limitation, the Orbitrap Astral mass spectrometer clearly outperforms the Orbitrap Exploris 480 mass spectrometer in terms of number of quantified proteins, sensitivity and quantitative accuracy, allowing for the investigation of heterogeneity of untreated cultured cells based on their size and cell cycle phase, as shown for A549 cells. Furthermore, the distinct separation we observed between TE-like cells and naive hPS cells highlights the potential of this SCP workflow for application to human blastocysts. It could clearly aid in exploring TE development mechanisms, blastocyst morphology regulation and the identification of molecular markers of high-quality blastocysts. Ultimately, this knowledge could contribute to our understanding of human blastocyst development and also potentially improve in vitro fertilization success rates.

In conclusion, we are convinced that our comprehensive workflow with optimized and improved parameters from sample preparation to data interpretation is a highly valuable contribution to evolve SCP to the next level in terms of sensitivity, reproducibility and quantitative performance, helping to transition the field of SCP from the developmental phase to a technique for the biologist's toolbox.

## Online content

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

## Methods

### Ethical approvals

The Wicell line H9 was used under agreement 20-WO-341 for a research program entitled 'Modeling early human development: establishing a stem cell-based 3D in vitro model of human blastocyst (blastoids)'. Blastoid generation was approved by the Commission for Science Ethics of the Austrian Academy of Sciences. This work did not exceed a developmental stage normally associated with 14 consecutive days in culture after fertilization even though this is not forbidden by the International Society for Stem Cell Research (ISSCR) Guidelines as far as embryo models are concerned. All experiments complied with all relevant guidelines and regulations, including the 2021 ISSCR guidelines that forbid the transfer of human blastoids into any uterus[35].

### Cultivation of A459, H460 and HeLa cells

Cells were cultured at 37 °C in a humidified atmosphere at 5% $CO_2$. A549 and H460 cells were grown in RPMI 1640 medium, and HeLa cells were grown in DMEM. Cell medium was supplemented with 10% fetal bovine serum (FBS; 10270, Fisher Scientific), 1× penicillin–streptomycin (P0781-100ML, Sigma-Aldrich), 100× L-glutamine (200 mM, 250030-024, Thermo Scientific) and 1 mM sodium pyruvate (for RPMI only; 4275, Sigma-Aldrich). Cells were grown to around 75% confluency before trypsinization with 0.05% Trypsin-EDTA (25300-054, Thermo Scientific), followed by washing three times with PBS. Cells were resuspended in PBS at a density of 200 cells per µl for isolation with the cellenONE (Cellenion).

### Sample preparation of A459, H460 and HeLa cells

A549, H460 and HeLa cell isolation, lysis and digestion were performed within a 384-well plate (Thermo Scientific Armadillo 45PCR Plate, 384 wells, 12657516) using the cellenONE robot as previously described[7]; the cellenONE robot was operated using its control software (v2.0-1143). Briefly, cells were sorted into wells containing 1 µl of master mix (0.2% DDM (D4641-500MG, Sigma-Aldrich), 100 mM triethylammonium bicarbonate (TEAB; 17902-500ML, Fluka Analytical), 3 ng µl⁻¹ trypsin (Trypsin Gold, V5280, Promega), 0.01% enhancer (ProteaseMAX, V2071, Promega) and 1% DMSO). For single-cell samples, cells were deposited into individual wells, whereas for 20- or 40-cell libraries, the respective cell number was sorted into a single well. Humidity and temperature were controlled at 50% and 15 °C during cell sorting. A549 and H460 cells were isolated at a given diameter of 15–30 µm and HeLa cells at 18–25 µm. The maximum elongation was set to 1.5. Cell lysis and protein digestion were performed at 50 °C and 85% relative humidity for 30 min before an additional 500 nl of 3 ng µl⁻¹ trypsin was added. After lysis and digestion, 3.5 µl of 0.1% TFA was added to the respective wells for quenching and storage at −20 °C. For LC–MS/MS analysis, samples were directly injected from the 384-well plate.

For benchmarking studies to proteoCHIP-based sample preparation, cells were isolated and prepared exactly as described in the manufacturer's manual. Standard (nonstringent) washing of chips was performed by sonication in methanol for 20 min, followed by extensive flushing with Milli-Q water and drying in the fume hood. A stringent washing protocol was established by Cellenion and is considered proprietary. In brief, 300 nl of master mix (0.2% DDM, 10 ng µl⁻¹ Trypsin Gold and 100 mM TEAB) was dispensed into each well of a LF48 proteoCHIP or an EVO 96 proteoCHIP (Cellenion) using the cellenONE robot. The LF48 chip was prefilled with 2 µl of hexadecane oil (H6703-100ML, Sigma-Aldrich) per well. ProteoCHIPs were cooled to 8 °C for cell isolation, which freezes the hexadecane (LF48). Cells were isolated as described above following stepwise heating to a final temperature of 50 °C at 85% relative humidity for lysis and digestion. Constant addition of water by the cellenONE robot kept the samples hydrated and at the approximate constant volume. After that, 3.5 µl of 0.1% TFA was added, and the proteoCHIP

LF48 was cooled on wet ice to refreeze the hexadecane and manually separated from the sample containing the aqueous phase by transferring the sample to individual wells of a low-binding 96-well PCR plate (EP0030129512, Twin.tec PCR Plate 96 LoBind, skirted). The EVO 96 proteoCHIP did not contain hexadecane and was directly placed on top of a 96-well PCR plate to transfer the sample by centrifugation (500g, 30 s).

### Naive hPS cell culturing

Naive PS cells were cultured on gelatin-coated plates with a feeder layer of gamma-irradiated mouse embryonic fibroblasts (MEFs). The coating and feeder layer preparation methods were as previously described[32]. Cells were cultured in PXGL medium composed of N2B27 basal medium supplemented with 1 µM PD0325901 (MedChemExpress, HY-10254), 1 µM XAV-939 (MedChemExpress, HY-15147), 2 µM Gö 6983 (MedChemExpress, HY-13689) and 10 ng ml⁻¹ human leukemia inhibitory factor (made in-house). Formulation of the N2B27 basal medium included 50% DMEM/F12, 50% neurobasal medium, N-2 and B-27 supplements, GlutaMAX supplement, nonessential amino acids and 100 µM 2-mercaptoethanol. Cells were maintained in a hypoxic chamber (5% $CO_2$, 5% $O_2$) and passaged every 3–4 days. All cell lines were routinely tested negative for mycoplasma.

### TE differentiation from naive hPS cells

Naive PS cells were dissociated using Accutase (Biozym, B423201) at 37 °C for 5 min. Gentle mechanical dissociation was performed using a pipette, followed by centrifugation to collect the cell pellet. The pellet was resuspended in PXGL medium supplemented with 10 µM Y-27632 (MedChemExpress, HY-10583). To exclude MEFs, the cell suspension was transferred onto gelatin-coated plates and incubated at 37 °C for 70 min. Cell counting and viability were determined using a Countess automated cell counter (Thermo Fisher Scientific) with trypan blue staining. Cells were seeded at a density of $1.2 × 10^5$ cells per well on Geltrex (0.5 µl cm⁻²)-coated six-well plates. Culturing was performed in a hypoxic chamber (5% $CO_2$, 5% $O_2$). On induction day 0, the medium was replaced with PDA83 medium comprising N2B27 basal medium, 1 µM PD0325901, 1 µM A83-01 (MedChemExpress, HY-10432) and 2% FBS. On day 1, 2 µM XMU-MP-1 was added. On days 2 and 3, the medium was switched to N2B27 with 2% FBS. On day 4, FACS was conducted using the following protocol.

### TE and hPS cell FACS sorting and sample preparation

TE-like cells and naive PS cells were dissociated using Accutase (B423201, Biozym) at 37 °C for 10 min and 5 min, respectively. Gentle mechanical dissociation was performed using a pipette. Cells were then stained with antibodies to TROP2 and SUSD2, respectively. For sorting TE-like cells, TROP2⁺ cells were selected, ensuring exclusion of nondifferentiated cells. Similarly, for naive hPS cell sorting, SUSD2⁺ cells were sorted into six-well plates to exclude MEFs. Under both conditions, dead cells were identified and excluded using DAPI staining. The FACS gating strategy is shown in Supplementary Figs. 14 and 15 for TE and hPS cells. Cells were sorted into 384-well plates (Thermo Scientific Armadillo 45PCR Plate, 384 well, 12657516), which already contained 1 µl of lysis buffer (100 mM TEAB (17902-500ML, Fluka Analytical), 0.2% DDM (D4641-500MG, Sigma-Aldrich), 3 ng µl⁻¹ trypsin (Trypsin Gold, V5280, Promega) and 0.01% enhancer (ProteaseMAX, V2071, Promega)) in each well. After sorting, well plates were moved to humidity- and temperature-controlled cellenONE chambers. Samples were incubated for 30 min at a humidity of 85% and a temperature of 50 °C. An additional 500 nl of 3 ng µl⁻¹ trypsin was added. Constant addition of water by the cellenONE robot kept the samples hydrated and at the approximate constant volume. After lysis and digestion, 3.5 µl of 0.1% TFA and 5% DMSO was added to the respective wells for quenching and storage at −20 °C. For LC–MS/MS analysis, samples were directly injected from the 384-well plate.

## Two-proteome mixes

HeLa (Thermo Scientific, Pierce HeLa Protein Digest Standard, 88328) and yeast (Promega, MS Compatible Yeast Protein Extract, Digest, *Saccharomyces cerevisiae*, 100 μg, V7461) were combined in 0.1% TFA at the following ratios and at a concentration of 250 pg μl$^{-1}$: HeLa:yeast = 200 pg:50 pg, 240 pg:10 pg and 150 pg:100 pg. One microliter (250 pg) of each two-proteome mix was injected to assess the limits of quantification at ultra-low-input levels.

## Diluted bulk digests

HeLa (Thermo Scientific, Pierce HeLa Protein Digest Standard, 88328) or K562 (Promega, Mass Spec-Compatible Human Protein Extract, V6951) cell peptides were dissolved in 0.1% TFA and 0.015% DDM (D4641-500MG, Sigma-Aldrich) at a concentration of 5 ng μl$^{-1}$ for injection into the LC–MS system.

## LC–MS analysis

Samples were analyzed using the Thermo Scientific Vanquish Neo UHPLC system. Thermo Tune software version 0.4 or higher was used to acquire data.

Peptides were separated on an Aurora Ultimate TS 25-cm nanoflow UHPLC column with an integrated emitter (Ion Optics) at 50 °C using direct injection mode.

Peptide separation was performed at 30–80 SPD with details provided in Supplementary Table 1 and with all single-cell measurements performed at 50 SPD. Fast sample loading was performed at a maximum pressure of $1.4 \times 10^8$ Pa and at a maximum flow rate of 1 μl min$^{-1}$.

For MS measuring, the Orbitrap Astral mass spectrometer and the Orbitrap Exploris 480 mass spectrometer (both Thermo Scientific) were coupled to the LC. Both instruments were equipped with a FAIMS Pro interface (Thermo Scientific) and an EASY-Spray source. Data recorded for reproducibility checks (Supplementary Fig. 1d) were recorded on a second Orbitrap Astral instrument equipped with a FAIMS Pro Duo interface. A compensation voltage of –48 V (Orbitrap Astral) or –50 V (Orbitrap Exploris 480) was used, if not indicated differently. An electrospray voltage of 1.85 kV was applied for ionization and adopted to slightly higher voltages for aged emitters to ensure spray stability.

On the Orbitrap Astral mass spectrometer, MS$^1$ spectra were recorded using the Orbitrap analyzer at a resolution of 240,000 from *m*/*z* 400 to 900 using an automated gate control (AGC) target of 500% and a maximum injection time of 100 ms. For MS$^2$ in DIA mode using the Astral analyzer, nonoverlapping isolation windows ranging from an *m*/*z* of 5 for 100 cells and 5- to 10-ng bulks to an *m*/*z* of 8 for 5-ng bulks to an *m*/*z* of 10 for 700-pg to 1-ng bulks and to an *m*/*z* of 20 for 1–40 cells, blanks and 50- to 500-pg bulks. A scan range from an *m*/*z* of 400 to 800 was chosen. The precursor accumulation time ranged from 10 ms for 100 cells and 5- to 10-ng bulks to 14 ms for 700-pg to 1-ng bulks to 40 ms for 250- to 500-pg bulks to 60 ms for 100- to 150-pg bulks, 1–40 cells and blanks and to a maximum of 80 ms for 50-pg bulks. The AGC target was set to 800%.

On the Orbitrap Exploris 480 mass spectrometer, the MS method was altered due to the lower speed and sensitivity of that instrument than the Orbitrap Astral mass spectrometer. However, comparable settings for high-throughput analysis as tested previously[13] were used with the following details: MS$^1$ spectra were recorded using the Orbitrap analyzer at a resolution of 120,000 from an *m*/*z* of 400 to 800 using an AGC target of 300%. For MS$^2$ in DIA mode, overlapping (*m*/*z* of 1) isolation windows of an *m*/*z* of 40 and a scan range from an *m*/*z* of 400 to 800 were chosen. The precursor accumulation time was set to a maximum of 118 ms, and the AGC target was set to 1,000%.

## Data analysis

All raw data were analyzed using Spectronaut (version 18.6.231227. 55695, Biognosys)[36]. For results marked as 'method evaluation',

DirectDIA+ was used in method evaluation mode without cross-normalization and with every raw file defined as a separate condition to ensure that Spectronaut treats each file as if they were analyzed individually. 'DirectDIA+' indicates that results were analyzed in Direct-DIA+ mode with replicates defined as the same condition in the search settings. For library searches, higher-input DIA recorded results (as given) were first analyzed in DirectDIA+ mode, and a library was created from those files using Spectronaut. Quantification was performed at the MS$^1$ level. Carbamidomethylation of cysteines as a static modification was removed for single-cell searches as no alkylation step was performed. Factory settings were used for all analyses and for library generation unless otherwise indicated. 'Optimized settings' indicates that searches were performed with a more stringent cutoff value of 0.01 (for precursor *q*-value cutoff, precursor posterior error probability (PEP) cutoff, protein *q*-value cutoff (experiment/run) and protein PEP cutoff) as described by Baker et al.[22]. Searches were performed against the human proteome (UniProt proteome UP000005640, reviewed, 20,408 protein entries, downloaded 4 August 2023) and the CRAPome[37] (118 protein entries). Searches for the two-proteome analysis were run against the same human database and the yeast proteome (UniProt proteome UP000002311, reviewed, 6,727 protein entries, downloaded 11 December 2023).

If not otherwise stated, FDR filtering was based on default settings of Spectronaut and at 1% at the protein level. For FDR checks, a decoy ('shuffled target') database was generated using the pyteomics[38] Python package in shuffled mode with fixed positions for arginine, proline and lysine to make sure that peptide mass and length distributions were the same as in a target database. A *C. elegans* (UniProtKB, downloaded 13 June 2024, 27,084 entities) database was also used as an entrapment database. The FDR was estimated using the following equation[39]:

$$\widehat{FDR} = \frac{d(1 + 1/r)}{t + d},$$

where *d* is the number of decoy proteins, *t* represents target proteins, and *r* represents the ratio between decoy and target databases.

## Postanalysis

**Human and yeast dataset.** Raw quantity values were used. All proteins with missing quantitative values were filtered out.

**Single-cell datasets.** For the statistical analysis of single-cell data, DirectDIA+ results were used. All proteins with missing values in more than 20 cells for A459 cells and 5 cells for the TE with hPS cell dataset were filtered out. For H460 and A549 cells (and blank files), dataset proteins with 20 or more missing values were filtered out. The rest of the missing values were substituted with intensities equal to the minimum in the dataset. Quantitative data were then transformed to log$_2$ format and normalized to normal distribution. The cell diameter data for PCA and UMAP clustering were taken from the cellenONE robot. PCA and UMAP clustering were performed using sklearn[40] (version 1.2.0) and umap[41] (version 0.5.6) Python packages. Cluster mapping of A549 and H460 cells and blanks was performed using the seaborn[42] (0.12.2) Python package.

To find statistical differences between hPS cells and TE cells, a Student's *t*-test for two independent samples was performed (scipy (version 1.11.4) Python package). The Benjamini–Hochberg FDR was used to correct *P* values for multiple comparisons (statsmodel (v. 0.14.1) Python package). The level of significance for corrected *P* values was set to 0.05. Proteins with fold change values (TE/hPS cells) greater than 1 were considered upregulated. GO analysis of upregulated proteins was performed using String-db.org[43] (version 12.0).

**Pseudobulk dataset of TE-like and hPS cells.** As a 'bulk' dataset, we considered 100-cell runs, both TE-like and hPS cells analyzed together (six runs in total). We filtered out all proteins that were missing in three

or more runs. The missing values were substituted with intensities equal to the minimum in the dataset. Quantitative data were then transformed to $\log_2$ format and normalized to normal distribution. GO analysis of bulk data was performed the same way as for single-cell data.

## Reporting summary

Further information on research design is available in the Nature Portfolio Reporting Summary linked to this article.

## Data availability

The MS raw proteomics data, Spectronaut search results and fasta files used have been deposited to the ProteomeXchange Consortium via the PRIDE[44] partner repository with the dataset identifier PXD049412. Source data are provided with this paper.

## Code availability

Data analysis of Spectronaut search results and figure plotting is available on GitHub at https://github.com/SimpleNumber/SCP_on_AstralMS and figshare at https://figshare.com/projects/SCP_on_AstralMS/226800 (ref. 45).

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

## Acknowledgements

This work was supported by the infrastructure funding fourth call 2022/01 (AT-SCP) of the Austrian Research Promotion Agency (FFG) and the project LS20-079 of the Vienna Science and Technology Fund (WWTF). This work was further funded by the P35045-B project (grant DOI 10.55776/P35045) of the Austrian Science Fund (FWF). This research was funded in whole, or in part, by the FWF. J.A.B.'s work was funded by FWF (grant DOI 10.55776/ESP497). We thank the laboratory of J. Penninger at Institute of Molecular Biotechnology for providing A549 cells. For the purpose of open access, we have applied a CC BY public copyright license to this publication.

## Author contributions

J.A.B. and M.M. conceptualized the study, designed and performed experiments, analyzed the data and wrote the paper. T.N.A., E.D. and B.D. performed experiments and designed experimental settings. J.S. and H.K. prepared TE and hPS cells. J.S., H.K., T.M.S. and N.R. interpreted TE and hPS cell data. P.P. contributed to the data analysis strategy. K.M. conceptualized the study and performed data analysis. All authors revised and agreed on the paper.

## Competing interests

T.N.A., E.D. and B.D. are employees of Thermo Fisher Scientific. The other authors declare no competing interests.

## Additional information

**Extended data** is available for this paper at https://doi.org/10.1038/s41592-024-02559-1.

**Correspondence and requests for materials** should be addressed to Julia A. Bubis, Karl Mechtler or Manuel Matzinger.

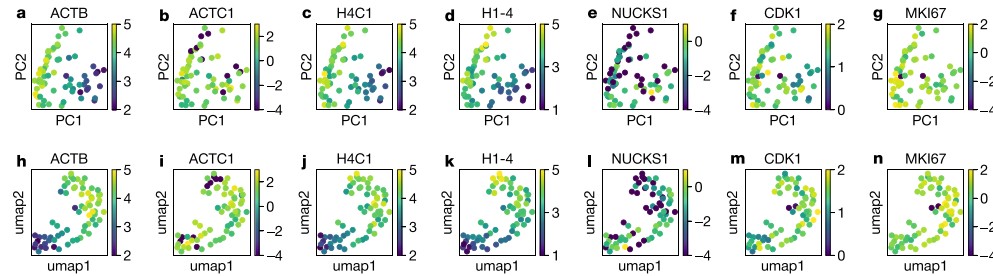

**Extended Data Fig. 1 | PCA & UMAP of A549 cells with relative abundancies of selected cell-cycle specific proteins color coded. a–n**, Individual A549 cells were prepared using the One-Pot workflow. Digested cells were analyzed at 50 SPD using the previously optimized settings for LC and MS on the Orbitrap Astral MS. The color scale bars depict the log2-transformed protein abundance. PCA (**a-g**) and UMAP (**h-n**) are based on protein quantities with each dot representing a cell and colors reflecting the relative abundance of the protein given.

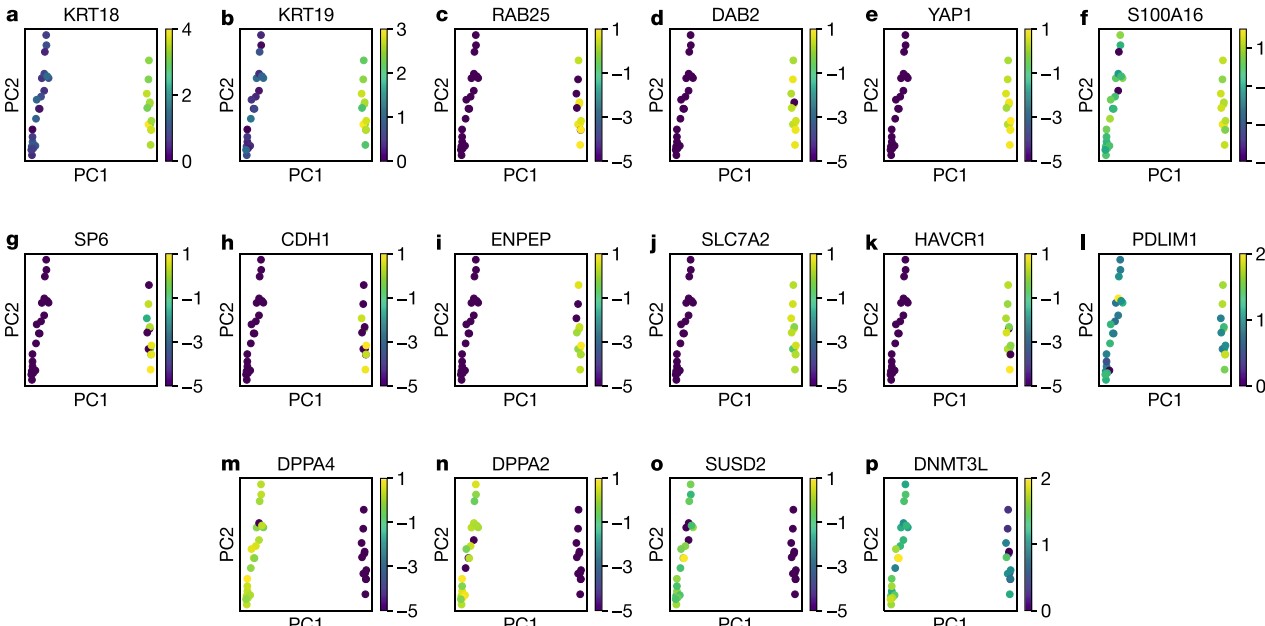

**Extended Data Fig. 2 | PCA & UMAP of hPSC & TE cells with relative abundancies of selected cell-type specific proteins color coded. a–p,** Individual hPSC and TE cells were prepared in a 384-well plate using the FACS. Digested cells were analyzed at 50 SPD using the previously optimized settings for LC and MS on the Orbitrap Astral MS. PCA on protein quantities with each dot representing a cell and colors reflecting the relative abundance of the protein given. The color bar corresponds to $\log_2$-transformed protein abundance. Selected markers of TE cells (**a-l**) and hPSC (**m-p**).

# Reporting Summary

## Statistics

For all statistical analyses, confirm that the following items are present in the figure legend, table legend, main text, or Methods section.

| n/a | Confirmed | |
|---|---|---|
| ☐ | ☒ | The exact sample size (*n*) for each experimental group/condition, given as a discrete number and unit of measurement |
| ☐ | ☒ | A statement on whether measurements were taken from distinct samples or whether the same sample was measured repeatedly |
| ☐ | ☒ | The statistical test(s) used AND whether they are one- or two-sided *Only common tests should be described solely by name; describe more complex techniques in the Methods section.* |
| ☐ | ☒ | A description of all covariates tested |
| ☐ | ☒ | A description of any assumptions or corrections, such as tests of normality and adjustment for multiple comparisons |
| ☐ | ☒ | A full description of the statistical parameters including central tendency (e.g. means) or other basic estimates (e.g. regression coefficient) AND variation (e.g. standard deviation) or associated estimates of uncertainty (e.g. confidence intervals) |
| ☐ | ☒ | For null hypothesis testing, the test statistic (e.g. *F*, *t*, *r*) with confidence intervals, effect sizes, degrees of freedom and *P* value noted *Give P values as exact values whenever suitable.* |
| ☒ | ☐ | For Bayesian analysis, information on the choice of priors and Markov chain Monte Carlo settings |
| ☐ | ☒ | For hierarchical and complex designs, identification of the appropriate level for tests and full reporting of outcomes |
| ☒ | ☐ | Estimates of effect sizes (e.g. Cohen's *d*, Pearson's *r*), indicating how they were calculated |

*Our web collection on statistics for biologists contains articles on many of the points above.*

## Software and code

Policy information about availability of computer code

| Data collection | Mass spectrometry data was acquired using the Orbitrap Astral MS or Orbitrap Exploris 480 MS, using Thermo Tune software (version: 0.4 or higher). Cell sizes were recorded on the cellenONE using its control software (v2.0-1143). |
|---|---|
| Data analysis | Proteomics data was analysed using Spectronaut (v. 18.6.231227.55695). For FDR checks a decoy ("shuffled target") database was generated using the Pyteomics (v. 4.6.2) Python package PCA and UMAP clustering were done using the sklearn (v. 1.2.0) and umap (v. 0.5.6) Python packages. t-test and FDR correction for multiple testing were performed using scipy (v. 1.11.4) and statsmodel (v. 0.14.1). Heatmap clustering was done using seaborn (v. 0.12.2) Python package. Python packages, respectively. Gene Ontology analysis was performed using string - db.org (v. 12.0). |

For manuscripts utilizing custom algorithms or software that are central to the research but not yet described in published literature, software must be made available to editors and reviewers. We strongly encourage code deposition in a community repository (e.g. GitHub). See the Nature Portfolio guidelines for submitting code & software for further information.

## Data

Policy information about availability of data

All manuscripts must include a data availability statement. This statement should provide the following information, where applicable:
- Accession codes, unique identifiers, or web links for publicly available datasets
- A description of any restrictions on data availability
- For clinical datasets or third party data, please ensure that the statement adheres to our policy

The mass spectrometry proteomics data along with FASTA files and result files have been deposited to the ProteomeXchange Consortium via the PRIDE partner repository with the dataset identifier PXD049412. Data analysis of Spectronaut search results and figure plotting are available on GitHub via link: https://github.com/SimpleNumber/SPC_on_AstralMS

## Human research participants

Policy information about studies involving human research participants and Sex and Gender in Research.

| | |
|---|---|
| Reporting on sex and gender | n.a. |
| Population characteristics | n.a. |
| Recruitment | n.a. |
| Ethics oversight | n.a. |

Note that full information on the approval of the study protocol must also be provided in the manuscript.

# Field-specific reporting

Please select the one below that is the best fit for your research. If you are not sure, read the appropriate sections before making your selection.

☒ Life sciences      ☐ Behavioural & social sciences      ☐ Ecological, evolutionary & environmental sciences

For a reference copy of the document with all sections, see nature.com/documents/nr-reporting-summary-flat.pdf

# Life sciences study design

All studies must disclose on these points even when the disclosure is negative.

| | |
|---|---|
| Sample size | To assess technical variability of the mass spectrometry measurements, each experiment was performed using at least three injection replicates. No sample size calculation was performed. As we saw a very low variance between these technical replicates aiming to check on the reproducibility of the instrument workflows themselves, we decided that no further replicates were needed.<br>For single-cell experiments 20 A549 and H460 cells where compared, for cell cycle investigations 66 A549 cells were analyzed and a total of 33 individual TE/hPSC cells was analyzed. For library generation and pseudo bulk analyses, 20, 40 or 100 cells were used as indicated in the main text. At least 3 blank controls were included in every single-cell study.<br>No sample size calculation was performed, but heterogeneity between cell types/cells was clearly visible with the number of measured replicates, indicating it is sufficient. We assume, even more cell replicates are always better, but are limited in measurement time. |
| Data exclusions | Single cells were excluded from downstream analysis if the number of identified proteins was less that 3 times number of proteins in the blanks. |
| Replication | All experiments were performed using 3 replicates of which all were successful |
| Randomization | For technical/methodological benchmarks no randomization was applied. Randomization/control of covariates is not applicable here as all 3 technical replicates are injected from the same vial and are hence identical. For single cell measurements, cells are selected randomly based on the design of the sample prep and filtered only for their size and circularity to exclude debris, multiple cells/well or dead cells. |
| Blinding | No blinding was performed or relevant to this work. A comprehensive overview of all LC-MS methods and sample preperation methods benchmarked in this study was needed to select the best perfoming option. Knowledge of cell size and type compared in this study was required to check if they were properly separated in our heterogenity studies (PCA, UMAP).<br>Technical replicates were used to assess precision and accuracy, and randomization was considered not relevant as we focused on improving and validating the capabilities of our MS but did not compare any sample treatments. Also no clinical sample were used in this study. |

# Reporting for specific materials, systems and methods

We require information from authors about some types of materials, experimental systems and methods used in many studies. Here, indicate whether each material, system or method listed is relevant to your study. If you are not sure if a list item applies to your research, read the appropriate section before selecting a response.

## Materials & experimental systems

| n/a | Involved in the study |
|-----|----------------------|
| ☐ | ☒ Antibodies |
| ☐ | ☒ Eukaryotic cell lines |
| ☒ | ☐ Palaeontology and archaeology |
| ☒ | ☐ Animals and other organisms |
| ☒ | ☐ Clinical data |
| ☒ | ☐ Dual use research of concern |

## Methods

| n/a | Involved in the study |
|-----|----------------------|
| ☒ | ☐ ChIP-seq |
| ☐ | ☒ Flow cytometry |
| ☒ | ☐ MRI-based neuroimaging |

## Antibodies

| | |
|---|---|
| Antibodies used | Human TROP-2 Alexa Fluor 488 conjugated antibody, R and D system FAB650G-100UG,Lot:1646351<br>Anti-SUSD2-PE Miltenyi Biotec Cat#: 130-117-682; Clone : W5C5, LOT: 5201104670 |
| Validation | Detects human TROP-2 in direct ELISAs and Western blots. In direct ELISAs, no cross-reactivity with recombinant human (rh) VCAM1 or rhICAM1 is observed.<br>Validation statements available from manufacturers:<br>anti-TROP2 (https://www.rndsystems.com/products/human-trop-2-antibody-77220_mab650)<br>anti-SUSD2 (https://www.miltenyibiotec.com/AT-en/products/susd2-antibody-anti-human-w5c5.html#gref) |

## Eukaryotic cell lines

Policy information about cell lines and Sex and Gender in Research

| | |
|---|---|
| Cell line source(s) | A549 and H460 cells were provided by the laboratory of Josef Penninger, their original commercial source is ATCC.<br>Human embryonic stem cells (Wicell line H9) reset to naive state were provided by the Laboratory of Yasuhiro Takashima |
| Authentication | H9 was included in single cell sequencing analysis to authenticate their identity in previous studies of Nicolas Rivron. We comfirm naïve state of the cells by assessing the expression of the naïve PSC marker SUSD2 through staining. Additionally, we routinely thaw a fresh stock vial to minimize the risk of cross-contamination with unintended cell lines and to prevent the accumulation of DNA mutations |
| Mycoplasma contamination | Cells were routinely tested for mycoplasma contamination. No contamination was detected |
| Commonly misidentified lines (See ICLAC register) | none |

## Flow Cytometry

### Plots

Confirm that:

☒ The axis labels state the marker and fluorochrome used (e.g. CD4-FITC).

☒ The axis scales are clearly visible. Include numbers along axes only for bottom left plot of group (a 'group' is an analysis of identical markers).

☒ All plots are contour plots with outliers or pseudocolor plots.

☒ A numerical value for number of cells or percentage (with statistics) is provided.

### Methodology

| | |
|---|---|
| Sample preparation | TE-like cells and naïve PSCs were dissociated using Accutase at 37°C for 10 minutes and 5 minutes, respectively. Gentle mechanical dissociation was performed using a pipette. Cells were then stained with antibodies against TROP2 and SUSD2, respectively. |
| Instrument | FACS Aria III (BD) |
| Software | DiVa version on the F02 is 9.0.1 |

| Cell population abundance | Abundance of the cell populations of interest was determined by the appropriate negative control and the purity of sorted population was assessed by the post sort analysis. |
| Gating strategy | FSC-A/SSC-A and SSC-H/SSC-W gates were applied to remove debris, and non-single cell aggregates respectively. Dead cells were excluded by using DAPI signal. |

☒ Tick this box to confirm that a figure exemplifying the gating strategy is provided in the Supplementary Information.

