## [Peer Review File · Nature Methods]

Challenging the Astral™ mass analyzer - up to 5300 proteins per single-cell at unseen quantitative accuracy to study cellular heterogeneity.

Corresponding Author: Dr Manuel Matzinger

Version 0:

Decision Letter:

10th May 2024

Dear Dr. Matzinger,

Your Article, "Challenging the Astral™ mass analyzer - up to 5300 proteins per single-cell at unseen quantitative accuracy to study cellular heterogeneity.", has now been seen by 2 reviewers. As you will see from their comments below, although the reviewers find your work of considerable potential interest, they have raised a number of concerns. We are interested in the possibility of publishing your paper in Nature Methods, but would like to consider your response to these concerns before we reach a final decision on publication.

We therefore invite you to revise your manuscript to address these concerns. We would like the revised version to address all the technical concerns that were raised as well as to make a stronger case for the advance over available methods.

Link Redacted

We hope to receive your revised paper within 6-8 weeks. If you cannot send it within this time, please let us know. In this event, we will still be happy to reconsider your paper at a later date so long as nothing similar has been accepted for publication at Nature Methods or published elsewhere.

OPEN SCIENCE REQUIREMENTS

REPORTING SUMMARY AND EDITORIAL POLICY CHECKLISTS

IMAGE INTEGRITY

DATA AVAILABILITY

All novel DNA and RNA sequencing data, protein sequences, genetic polymorphisms, linked genotype and phenotype data, gene expression data, macromolecular structures, and proteomics data must be deposited in a publicly accessible database, and accession codes and associated hyperlinks must be provided in the "Data Availability" section.

MATERIALS AVAILABILITY

SUPPLEMENTARY PROTOCOL

To help facilitate reproducibility and uptake of your method, we ask you to prepare a step-by-step Supplementary Protocol for the method described in this paper. We [encourage authors to share their step-by-step experimental protocols](https://www.nature.com/nature-research/editorial-policies/reporting-standards#protocols) on a protocol sharing platform of their choice and report the protocol DOI in the reference list. Nature Portfolio 's Protocol Exchange is a free-to-use and open resource for protocols; protocols deposited in Protocol Exchange are citable and can be linked from the published article. More details can found at www.nature.com/protocolexchange/about.

ORCID

Sincerely,
Arunima

Arunima Singh, Ph.D.
Senior Editor
Nature Methods

Reviewers' Comments:

Reviewer #1:

Remarks to the Author:

Bubis and colleagues benchmark Orbitrap Astral for single cell analyses. This work achieves a quite remarkable proteomic depth, demonstrating that comprehensive single cell proteomics via label-free DIA is possible with up to 80 samples/day throughput. Another highlight of the manuscript is mixed species quantification benchmarks. To my knowledge, this is the first time these are carried out using single cell-level sample amounts, showing comprehensive quantitative performance.

I have a number of comments concerning the technical side of the paper.

1. FDR benchmarks. Nice to see this shown, given that concerns were raised by some colleagues specifically about the Astral data. I would also suggest a number of adjustments.

First, the protein-level FDR can be calculated as roughly $(\text{decoy hits} / \text{all hits}) * [\# \text{ proteins considered} - \# \text{ proteins correctly identified}] / [\# \text{ number of decoy proteins}]$. That is if there are, say, 6k true IDs, searching against the 21k-genes uniport proteome and the same number of decoys, and 60 (1%) of IDs called are decoys, then the effective FDR estimate will be $(60 / 6000) * [21k * 2 - 6k] / 21k = 1.7\%$. For this work, it's fine if the software has a bit elevated FDR though, sufficient to demonstrate that it is reasonable.

Second, this calculation can only be applied in the Method Evaluation mode, as benchmarking FDR in directDIA/MBR would require a setup wherein a lysate from a particular species is spiked into some samples but not other samples, thus allowing to estimate the false id transfer between runs. This however in my opinion would be unnecessary for this work.

Finally, this only works for proteins. To evaluate the FDR for precursors, one would need to disable any kind of protein-level filtering in the processing software. Otherwise this indeed leads to strangely low values like just 0.1% decoy precursor IDs. I would like to note that this is just a property of an entrapment benchmark, that is protein FDR filtering does not really reduce the precursor FDR within selected proteins at all, it just affects the benchmark.

I think using e.g. a C.elegans proteome as the decoy proteome, rather than a shuffled database, would probably be preferable, as using a shuffled database for DIA is not something established in literature. However, it is also fine as is.

2. The DIA-based library approach shows performance comparable to directDIA on actual single cells but seems to outperform directDIA significantly on peptide standards - this is very much expected. However, one concern here is that the FDR with the library-based method might be different (less conservative) – directly this can only be measured if the library is created using two-species samples. This has been done for old versions of the popular software tools (PMID: 35551187), showing severe FDR inflation for all of them, when using DIA-based libs. At least in DIA-NN this has been fixed since then, however this is something that always need to be considered, especially in such a challenging for the software scenario as the analysis of single cell-level amounts. In the case of the present manuscript, it is possible to do the following check, which I suggest the authors include to make sure there is no FDR inflation with the library. Please try directDIA including both the 250pg runs and 10ng runs: in this case the number of IDs for 250pg should be about the same as with the library. If it is less, this likely means some FDR inflation with the library. Even if this is the case, it will not reduce the value of the paper.

3. Very interesting to see the advantage of FAIMS for single cell proteomics, considering that it seems to lower the signals considerably. Here I would suggest to also add a plot that shows the distribution of precursor signals for precursors detected both with and without FAIMS – allowing to evaluate the degree of signal loss with FAIMS.

4. Is there any difference in the degree of carry over between different gradient lengths?

5. Please also add MS2-level mixed-species quantification benchmarks in Supplementary.

Vadim Demichev

Reviewer #2:

Remarks to the Author:

This study by Matzinger et al. utilized the one-pot 384 well workflow with a working volume of 1 μ L. They combined this with the commercial cell sorting device cellenONE (Cellenion, Denmark) and the newly launched mass spectrometry instrument Orbitrap Astral (Thermo, San Jose), successfully identify over 5000 proteins under specific conditions, including database search settings, LC gradients, and cell sizes. The authors also benchmarked multiple data analysis strategies to achieve accurate quantification at single-cell level inputs. Finally, they applied the method to distinguish between in vitro analogs of two human blastocyst: TE-like cell and naïve hPSCs. Overall, the study was well-executed, with a clear logical flow, and the manuscript was generally well-written.

However, there are several major issues in the manuscript. Based on the following comments, we believe that this manuscript is more suitable for publication in the journal of Analytical Chemistry.

Major Points:

1. The manuscript lacks detailed comparison of three sample preparation workflows. The authors have not effectively demonstrated the advantages of the on-pot method when compared to other techniques, nor have they provided data to support the claim of outcome in sample transfer step. It is imperative that the authors present detailed data illustrating the benefits of the tip in preventing sample loss

2. The fundamental principle of “minimizing surface adsorption losses” hinges on reducing surface contact, thereby minimizing surface adsorption. In the supplemental figure 3, the difference seems to focus on the containment. The authors need to explain the source of excessive containment in ProteoCHIP LF48 and Evo96 that not mentioned in other articles.

3. The authors have devoted significant space to comparing and optimizing experimental conditions like gradient and FAIMS pro for standard samples. While some aspects have been previously reported, there is insufficient analysis of truly single-cell proteomic data. For instance, the data completeness and coefficient of variation in different single cells and a comparison with bulk proteomic data is needed to highlight the advantages of single-cell analysis.

4. The significant advancement in single-cell sample pretreatment system is the OAD chip, a nanoliter droplet system (SODA) combined with shotgun proteomics, developed by Q. Fang and CCL Wong’s group (Li et al. Analytical Chemistry, 2018). This innovation has notably minimized the surface adsorption and significantly enhance the protein identification. It is essential to recognize this as the original work of NanoPOTS (reference 4 in the current manuscript), which should be duly cited.

While this work serves as an applied study for optimizing parameters across various commercial devices (including cell sorting, LC and MS), it lacks universality. To prove the method’s true innovation and outstanding results, the authors need to further discuss the single-cell proteomic data.

Other Points:

1. It is widely understood that using Match-Between-Run (MBR) or a protein carrier can significantly boost protein ID, making it not a true SC analysis. The authors should explicitly indicate which method is utilized for the biological analyses, such as discussions on A549 cell heterogeneity and cell cycle, as well as research on the discovery of TE cell biomarkers.

2. Given that the protein amount of a single mammalian cell (like HeLa cell) is as low as 150 pg, and accounting for inevitable losses in sample preparation including protein digestion, the authors could utilize 100 pg standard sample to evaluate the workflow method to enable true single-cell proteomics.

3. The authors have overstated the protein identification results and should temper their claims, especially in the Abstract,

Introduction and Discussion sections. Protein identification levels appear to fluctuate due to the use of carriers, cell size selection, and different gradients.

Version 1:

Decision Letter:

Our ref: NMETH-A55495A

21st Aug 2024

Dear Dr. Matzinger,

Thank you for submitting your revised manuscript "Challenging the Astral™ mass analyzer - up to 5300 proteins per single-cell at unseen quantitative accuracy to study cellular heterogeneity." (NMETH-A55495A). It has now been seen by the original referees and their comments are below. The reviewers find that the paper has improved in revision, and therefore we'll be happy in principle to publish it in Nature Methods, pending minor revisions to satisfy the referees' final requests (as outlined in your revision plan, and approved by the reviewer) and to comply with our editorial and formatting guidelines.

TRANSPARENT PEER REVIEW

ORCID

Sincerely,
Arunima

Arunima Singh, Ph.D.
Senior Editor
Nature Methods

Reviewer #1 (Remarks to the Author):

The authors have addressed most of my questions. I only have minor comments left:

- The FDR formula used introduces a bias. Here I would like to refer to the recent excellent work by Bo Wen, Uri Keich and colleagues <https://doi.org/10.1101/2024.06.01.596967>, which details on why this formula should not be used. I would suggest to replace the FDR formula with the one I suggested in my review or with formula (1) in the above preprint. The true FDR in the benchmarks shown in the present manuscript is ~1.6-1.8%, if a correct formula is used, which is OK here.
- I don't think lower numbers in directDIA+ mode (page 4:128) indicate more stringent FDR control or improved data quality, but they could rather reflect the presence of global FDR control on top of run-specific one – I can only speculate about this, I am not familiar with the specifications of Spectronaut settings. But this alone cannot on itself be used as evidence about FDR or quality.
- Just to confirm, Supplementary Figure 15 is generated using non-normalised intensities? Please also indicate what sample & amount is being measured.

- Supplementary Figure 19: median 75% CV is huge even for single cells. Please check that this refers to normalised data and maybe check intensity histograms for ubiquitous proteins in R to verify that Spectronaut normalised things correctly here (no need to include this in the manuscript though, if all checks out).

Reviewer #2 (Remarks to the Author):

After carefully review the revised manuscript, I think the authors have addressed all the reviewers' concerns and make a good improvement to the manuscript. Therefore I agree that this manuscript should also be published on the Nature Method.

Version 2:

Decision Letter:

6th Nov 2024

Dear Manuel,

I am pleased to inform you that your Article, "Challenging the Astral™ mass analyzer - up to 5300 proteins per single-cell at unseen quantitative accuracy to study cellular heterogeneity.", has now been accepted for publication in Nature Methods. The received and accepted dates will be February 22, 2024 and November 6, 2024. This note is intended to let you know what to expect from us over the next month or so, and to let you know where to address any further questions.

Over the next few weeks, your paper will be copyedited to ensure that it conforms to Nature Methods style. Once your paper is typeset, you will receive an email with a link to choose the appropriate publishing options for your paper and our Author Services team will be in touch regarding any additional information that may be required. It is extremely important that you let us know now whether you will be difficult to contact over the next month. If this is the case, we ask that you send us the contact information (email, phone and fax) of someone who will be able to check the proofs and deal with any last-minute problems.

Please note that *Nature Methods* is a Transformative Journal (TJ). Authors may publish their research with us through the traditional subscription access route or make their paper immediately open access through payment of an article-processing charge (APC). Authors will not be required to make a final decision about access to their article until it has been accepted. [Find out more about Transformative Journals](https://www.springernature.com/gp/open-research/transformative-journals)

If you are active on Twitter/X, please e-mail me your and your coauthors' handles so that we may tag you when the paper is

published.

Best regards,
Arunima

Arunima Singh, Ph.D.
Senior Editor
Nature Methods

** Visit the Springer Nature Editorial and Publishing website at http://editorial-jobs.springernature.com?utm_source=ejP_NMeth_email&utm_medium=ejP_NMeth_email&utm_campaign=ejp_Nmeth for more information about our career opportunities. If you have any questions please click [here](mailto:editorial.publishing.jobs@springernature.com).**

Reviewers' Comments:

Reviewer #1:

Remarks to the Author:

Bubis and colleagues benchmark Orbitrap Astral for single cell analyses. This work achieves a quite remarkable proteomic depth, demonstrating that comprehensive single cell proteomics via label-free DIA is possible with up to 80 samples/day throughput. Another highlight of the manuscript is mixed species quantification benchmarks. To my knowledge, this is the first time these are carried out using single cell-level sample amounts, showing comprehensive quantitative performance.

I have a number of comments concerning the technical side of the paper.

We thank Dr. Demichev for carefully going through our manuscript and the encouraging and positive feedback. We considered all concerns and replied along the lines.

1. FDR benchmarks. Nice to see this shown, given that concerns were raised by some colleagues specifically about the Astral data. I would also suggest a number of adjustments.

*First, the protein-level FDR can be calculated as roughly (decoy hits / all hits) * [# proteins considered - # proteins correctly identified] / [# number of decoy proteins]. That is if there are, say, 6k true IDs, searching against the 21k-genes uniport proteome and the same number of decoys, and 60 (1%) of IDs called are decoys, then the effective FDR estimate will be (60 / 6000) * [21k * 2 - 6k] / 21k = 1.7%. For this work, it's fine if the software has a bit elevated FDR though, sufficient to demonstrate that it is reasonable.*

Second, this calculation can only be applied in the Method Evaluation mode, as benchmarking FDR in directDIA/MBR would require a setup wherein a lysate from a particular species is spiked into some samples but not other samples, thus allowing to estimate the false id transfer between runs. This however in my opinion would be unnecessary for this work.

Finally, this only works for proteins. To evaluate the FDR for precursors, one would need to disable any kind of protein-level filtering in the processing software. Otherwise this indeed leads to strangely low values like just 0.1% decoy precursor IDs. I would like to note that this is just a property of an entrapment benchmark, that is protein FDR filtering does not really reduce the precursor FDR within selected proteins at all, it just affects the benchmark.

I think using e.g. a C.elegans proteome as the decoy proteome, rather than a shuffled database, would probably be preferable, as using a shuffled database for DIA is not something established in literature. However, it is also fine as is.

First and foremost, we would like to extend our sincere apologies for not providing a detailed explanation of the FDR estimation in our article. The formula we used is as follows:

$$\widehat{FDR} = \frac{d}{t \cdot r}$$

where d is the number of identified decoys, t is the number of identified targets, and r is the ratio between the decoy and target databases.

We acknowledge that these calculations are applicable only to the Method Evaluation data, and thus, we have removed the DirectDIA data accordingly. Additionally, we have included a *C. elegans* database for entrapment, which resulted in a similar percentage (below 1% FDR) of false positives when using both shuffled and *C. elegans* databases. We agree that this estimation is accurate only at the protein level and not for precursors. To avoid any confusion, we have excluded this part.

Action taken: We updated Figure 3, now including *C. elegans* as entrapment database in addition to the shuffled database.

2. The DIA-based library approach shows performance comparable to directDIA on actual single cells but seems to outperform directDIA significantly on peptide standards - this is very much expected. However, one concern here is that the FDR with the library-based method might be different (less conservative) – directly this can only be measured if the library is created using two-species samples. This has been done for old versions of the popular software tools (PMID: 35551187), showing severe FDR inflation for all of them, when using DIA-based libs. At least in DIA-NN this has been fixed since then, however this is something that always need to be considered, especially in such a challenging for the software scenario as the analysis of single cell-level amounts. In the case of the present manuscript, it is possible to do the following check, which I suggest the authors include to make sure there is no FDR inflation with the library. Please try directDIA including both the 250pg runs and 10ng runs: in this case the number of IDs for 250pg should be about the same as with the library. If it is less, this likely means some FDR inflation with the library. Even if this is the case, it will not reduce the value of the paper.

Thank you for this suggestion. We did DirectDIA searches using both 250pg and 10ng HeLa files. Here is the figure with # of PGs and Precursors identified:

Point-to-point figure 1: Identified Protein Groups and Precursors in HeLa 250pg runs searched against library from 10ng HeLa runs and searched in DirectDIA mode together with 10ng HeLa runs.

As one can see the PG numbers are similar to each other (difference ~0,4%), and on Precursor level the difference is ~5%, suggesting that FDR inflation is no problem in the used Spectronaut version (18.6).

Action taken: We updated the discussion of FDR in the revised manuscript and added Point-to-point figure 1 as additional panel to Supplemental Figure 1.

3. Very interesting to see the advantage of FAIMS for single cell proteomics, considering that it seems to lower the signals considerably. Here I would suggest to also add a plot that shows the distribution of precursor signals for precursors detected both with and without FAIMS – allowing to evaluate the degree of signal loss with FAIMS.

Thank you for this comment, it is a valuable addition to our investigation of FAIMS interface usage. We took a closer look at the intersection of found peptides from “CV-48” and “no FAIMS” runs (15663 entities) and observed intensity differences. The general log₂-intensity distribution has a similar shape, however, data with FAIMS has significantly fewer missing or zero values (Point-to-point figure 2). We pairwise compared the intensity of the peptides, Point-to-point figure 2b shows the distribution of log-transformed peptide abundance ratios in CV-48 runs to noFAIMS runs. This distribution is bimodal. The major part of peptides has intensity shifts in both directions with a mean value around zero and for a small group of peptides (~1000 peptides) a significant increase in abundance values is visible.

Point-to-point figure 2: a - Peptide abundance distribution in runs with CV-48V and no FAIMS interface runs. Peptide abundances were calculated using MS1 intensity. b- distribution of log₂-transformed peptide abundance ratio in CV-48 and noFAIMS runs.

Action taken: We included Point-to-point figure 2 to the manuscript as Supplemental Figure 15 and added a respective short description to the FAIMS section of our revised manuscript.

4. Is there any difference in the degree of carry over between different gradient lengths?

Thank you for this very important question. Carry over is potentially highly problematic as it would bias cell-cell differentiation. However, with the low inputs used and in direct injection mode, we do not assume carry-over being an issue in this study.

To investigate a potential influence of the gradient length we recorded new data using 250 pg HeLa and subsequent wash runs. This data was recorded at 30, 50 or 80 SPD throughput to cover the entire range used in this study. In method evaluation mode data analysis of wash-runs failed in Spectronaut 18 as not enough data for calibration was available in those files. Hence, we consider those as zero identification and no carryover (Point-to-point figure 3 A). When allowing for matching in each group of 5 replicates, results are obtained and show a clear drop of more than two orders of magnitude in total protein quantity (Point-to-point figure 3 C). Number wise, the first wash contains as little as 0.94%, 1.6% and 1.3% of the number of precursors seen in the previous 250 pg injection for 30, 50 and 80 SPD respectively (Point-to-point figure 3 B). Numbers go further down for subsequent washes indicating a low level of carryover that is washed out with time. However, as numbers are that low compared to sample injections and no accumulation of sample (constant numbers from first to fifth 250 pg injection) seems visible we conclude carryover as not problematic for all tested gradient lengths.

We additionally visually checked TICs from wash- and blank-runs for carryover from our preferred and predominantly used 50SPD method. This method was also used for all single cell measurements. Close to 0 proteins/precursors were identified in those wash runs (injection of 0.1% TFA) and visual inspection of base peak chromatograms shows that nothing but noise is visible in our wash runs, indicating low to no carryover (Point-to-point figure 4). In blank runs, containing everything but a cell, the dominant trypsin peaks are visible as expected but peptide peaks are missing.

Point-to-point figure 3: Identified protein groups and precursors from 250 pg HeLa diluted bulk and washruns, shown in the sequence as measured on the 25 cm IonOpticks column in direct injection mode and at a throughput as indicated. Analyzed in method evaluation mode without matching (A) or using directDIA+ with matching (B) in Spectronaut 18. Panel C shows the non-normalized sum of all protein group quantities reported from the directDIA+ analysis.

Point-to-point figure 4: Examples of base peak chromatograms for wash-runs (injection of 0.1% TFA) after injection of 250 pg diluted bulk digest (A), blank-runs (containing everything a sample contains but no cell), or a single cell sample (B). All measured using the 50SPD gradient on the Astral as used as preferred method in the manuscript.

Action taken: We included Point-to-point figure 3 to the supplement of the revised manuscript and added a short discussion on the level of carryover to expect in our results section.

5. Please also add MS2-level mixed-species quantification benchmarks in Supplementary.

We updated Supplementary Figures with MS2-based quantitation for HeLa + Yeast mixes. MS2 level quantitation has wider distribution for ratios in comparison with MS1-based quantitation.

Action taken: We added MS2 level benchmarks as requested, now Supplemental Figure 6 in the revised manuscript.

Vadim Demichev

Reviewer #2:

Remarks to the Author:

This study by Matzinger et al. utilized the one-pot 384 well workflow with a working volume of 1 μ L. They combined this with the commercial cell sorting device cellenONE (Cellenion, Denmark) and the newly launched mass spectrometry instrument Orbitrap Astral (Thermo, San Jose), successfully identify over 5000 proteins under specific conditions, including database search settings, LC gradients, and cell sizes. The authors also benchmarked multiple data analysis strategies to achieve accurate quantification at single-cell level inputs. Finally, they applied the method to distinguish between in vitro analogs of two human blastocyst: TE-like cell and naïve hPSCs. Overall, the study was well-executed, with a clear logical flow, and the manuscript was generally well-written. However, there are several major issues in the manuscript. Based on the following comments, we believe that this manuscript is more suitable for publication in the journal of Analytical Chemistry.

We thank the reviewer for carefully going through our manuscript and the constructive feedback highlighting its strengths. We considered all concerns and replied along the lines. We added a third single cell dataset to not only demonstrate the biological applicability but also reproducibility of our workflow. We are convinced that our revised manuscript is of substantially improved quality and highly valuable for the readership of Nature Methods due to the outstanding proteomic coverage at maintained quantitative accuracy at single cell level.

Major Points:

1. The manuscript lacks detailed comparison of three sample preparation workflows. The authors have not effectively demonstrated the advantages of the on-pot method when compared to other techniques, nor have they provided data to support the claim of outcome in sample transfer step. It is imperative that the authors present detailed data illustrating the benefits of the tip in preventing sample loss

We fully agree with the reviewer's opinion to clearly present data to back up all statements given in our manuscript. We therefore show in Supplemental Figure 3 a comparison of the One-Pot workflow to either the LF48 or the Evo96 chip-based workflow. There we found no advantage in terms of ID numbers and also in terms of useability (e.g. hexadecane needs to be removed and samples need to be transferred manually from the LF48 chip which obviously seems less convenient compared to a one-pot strategy).

We already reported sample loss by transfer in our previously published work and add the respective figure here for convenience (Point-to-point figure 5).

Since both chip-based workflows require a transfer step we assume a similar effect. As shown in Supplemental Figure 3, we could confirm this in our current data as well. Both the Evo96 chip and the 384 well plate used have a polypropylene surface in contact with the sample. Both workflows were performed from the same cell-batch and using DDM for lysis and the same batch of trypsin for digestion within the same cellenONE instrument.

We therefore assume that differences in proteomic depth occur mainly from the sample transfer step itself.

For better visualization we normalized the data of Supplemental Figure 3 to the respective result obtained when using the One Pot workflow. As shown in Point-to-point figure 6, the One Pot workflow thereby clearly outperforms both other workflows that were tested in this study.

Point-to-point figure 5: Effect of sample transfer after preparation of single HeLa cells in a 384 well plate using the cellenONE followed by injection from that plate or a single pipetting step to transfer the sample to the indicated tube. Measured on an Exploris480 in DDA mode. Data not part of this manuscript but from our previously published work (Matzinger & Müller et. al., Anal. Chem., 2023)

Point-to-point figure 6: Direct comparison of all 3 tested single cell sample preparation workflows enabled by normalization of identified protein groups for each workflow to the number of identified protein groups using the One Pot workflow of the same batch. This graph uses data from Supplemental Figure 3.

Action taken: We adopted the results section in our manuscript respectively to make all details of the 3 sample preparation workflows clearer to the readership.

2. The fundamental principle of “minimizing surface adsorption losses” hinges on reducing surface contact, thereby minimizing surface adsorption. In the supplemental figure 3, the difference seems to focus on the containment. The

authors need to explain the source of excessive containment in ProteoCHIP LF48 and Evo96 that not mentioned in other articles.

We fully agree with the reviewer's opinion: It is essential to minimize surface contact. This was indeed also our initial thought to even test the LF48 and Evo96 chips for this work rather than sticking to the -in house developed- 384 well one-pot workflow. The chip-based workflows should benefit from reduced surface losses thanks to the reduced volume of only 300 nL during digestion. This at the same time allows for higher concentrations of trypsin, which we expected to be advantageous as well. As detailed in our reply to the previous question, this expectation did not turn true, likely due to the sample transfer step that brings the peptide mix into contact with a fresh surface.

We further observe (as detailed above and shown in Supplemental Figure 3) more protein IDs when performing the workflow on washed but already used chips. Since those chips are quite expensive, multiple usage seems a valuable option for the community. However, it bears the risk of contamination. For that reason, we included blank runs, processed on the same chip but without a cell, as negative control. We show that the applied washing protocol has a huge impact on the level of contamination.

We further fully agree with the reviewer that most other works do not mention contamination and this is exactly the reason why we decided to do so! We think that background proteins might be highly problematic, and the information of the background control/blank runs is highly important to estimate the quality of the sample preparation. We therefore emphasize the importance of including negative controls in our manuscript and show for the first time to what extent this can be problematic for such ultra-low input samples if not taken care of.

Action taken: We adopted the results section in our manuscript to make clear that the extensive background results from re-used chips – not visible anymore after stringent washing or in the case of (fresh) 384 well plates used.

3. The authors have devoted significant space to comparing and optimizing experimental conditions like gradient and FAIMS pro for standard samples. While some aspects have been previously reported, there is insufficient analysis of truly single-cell proteomic data. For instance, the data completeness and coefficient of variation in different single cells and a comparison with bulk proteomic data is needed to highlight the advantages of single-cell analysis.

The authors thank for this comment. Indeed, we did not emphasize enough the characteristics of the acquired data sets. For this reason, for A549 cells, we now took a closer look at the data completeness (Point-to-point Figure 7) and CV (Point-to-point Figure 8) of our single cell data. As expected, cells differ from each other quite a lot even on identification level of protein groups (PG), so we can see that data completeness drops significantly with the number of files reaching only 1053 out of 5805 PG (~18%) identified in all 66 files. Also, there is a slight dependence on cell size, for 15-20 μm range of cell diameter we have slightly lowered data completeness 19 files in comparison with two other groups (20-25 μm and 25-30 μm , Point-to-point Figure 7 panels B-D). As expected,

mainly due to biological heterogeneity, CV distributions for this dataset also looks completely different from HeLa or K562 dilution series, where CV is below 10%. The median CV distribution for single-cell runs is 75%, while for stock runs of 20 and 40 cells the median 17% and 16%, respectively. This difference in CV aligns with our assumption that cells have a unique protein abundance distribution, which results in a high CV among single cells. In 20 and 40 cells samples the CV is much lower, since we average among 20 or 40 cells, however it is not close to HeLa or K562 dilution series. Several factors might be the issue:

(1) 20 or 40 cells is not enough to average all biological differences between cells; (2) for real datasets the sample lysis and digestion bring some fluctuations.

Point-to-point figure 7: a: Data completeness for all single-cell files of the A549 cell line, and b,c,d – data completeness for a subgroup based on cell size. For data completeness graphs was used DirectDIA+ search results of Spectronaut 18.

Point-to-point figure 8: CV distribution of PG quantitation values for single-cell data (A), 20 cells (B), 40 cells (C) of A549 cell line. Blue line indicates the median of the distribution. For CV graph was used DirectDIA+ search results of Spectronaut 18.

We calculated the characteristics for the TE and hPSC dataset as well (Point-to-point Figures 9 and 10). 53% of hPSC and 60 % of TE-like proteins were found across all files. The median CV of 44% is lower than for A549 cells. There might be several reasons for this: first, the dataset of TE and hPSC is smaller than A549 dataset; second, for A549 cells the total range of cell diameter is 15-30, which is wider and might be a plausible cause of that higher variability.

Point-to-point figure 9: Data completeness at protein group level of single-cell runs of TE (A) and hPSC (B). DirectDIA+ search results of Spectronaut 18 were used for the figure.

Point-to-point Figure 10: CV for TE and hPSC dataset. A: TE single-cell runs; B: TE 100-cell runs; C: hPSC single-cell runs; D: hPSC 100-cell runs. Protein abundance quantitation is based on MS1 intensities. DirectDIA+ search results of Spectronaut 18 were used for the figure.

For TE and hPSC we also compared quantitative analysis on both levels bulk (100 cell runs) and single-cell level. For marker proteins we saw the same figure trends (Supplementary Figure 8 and 11). In bulk data more pathways are relevant to the studying system. If we compare the pathways that were also found in single-cell data, we observe a similar number of identified proteins in bulk and single-cell data, however p-values in bulk data are lower.

Action taken: We added the technical characteristics of the aforementioned single-cell datasets and created Supplementary Figures 18-21 and compared them in the main text to the analysis of pseudo-bulk (100 cells) data of TE and hPSC (Supplementary Figures 10, 12).

4. The significant advancement in single-cell sample pretreatment system is the OAD chip, a nanoliter droplet system (SODA) combined with shotgun proteomics, developed by Q. Fang and CCL Wong's group (Li et al. Analytical Chemistry, 2018). This innovation has notably minimized the surface adsorption and significantly enhance the protein identification. It is essential to recognize this as the original work of NanoPOTS (reference 4 in the current manuscript), which should be duly cited.

Action taken: We added the citation of the OAD work to the list of cited examples on sample preparation workflows for single cell proteomics, which is reference 9 in the revised manuscript.

While this work serves as an applied study for optimizing parameters across various commercial devices (including cell sorting, LC and MS), it lacks universality. To prove the

method's true innovation and outstanding results, the authors need to further discuss the single-cell proteomic data.

We believe that our study is of great value to the community and that it is universally applicable especially because it uses only commercially available parts accessible to everyone without the need to spend months or years to implement specific home-made robots for sample preparation or pack own columns. Still in the shown combination and with the applied optimization in gradient length, FAIMS condition, sample preparation selection etc. the study reflects a novel and unique work yielding excellent proteome coverage while maintaining a low FDR of 1% at protein level and an excellent quantification performance (given the ultra-low input).

Furthermore, the following outstanding results, were, to the best of the authors knowledge, shown for the first time in our study:

- We are the first to assess quantitative performance at single cell input level (=250 pg in this study, with as little as 10 pg yeast inside the sample in the most extreme case)
- first to visualize cellular heterogeneity dependent on size/cell cycle only but without any cell treatment or pre-differentiation
- We are the first to describe hPSC and TE-like cell proteomes at single-cell level.
- We reproducibly observe an extremely high proteomic coverage, not seen before by the authors
- We are the first to compare sample preparation techniques which are easily commercially accessible using the cellenONE, serving as selection help for (new and old) users of that increasingly popular instrument.
- We are the first to report the importance to include blank runs into a study aiming to estimate the quality of sample preparation

We are convinced that these key points highlight the novelty, universal applicability, and significance of this study.

We further included an additional single cell proteomic dataset (Figure 6 in the revised manuscript) reproducing and even slightly improving our average proteome coverage.

Other Points:

1. It is widely understood that using Match-Between-Run (MBR) or a protein carrier can significantly boost protein ID, making it not a true SC analysis. The authors should explicitly indicate which method is utilized for the biological analyses, such as discussions on A549 cell heterogeneity and cell cycle, as well as research on the discovery of TE cell biomarkers.

For all quantitative analysis, namely PCA, UMAP, t-test, GO analysis we used DirectDIA+ results. Because in PCA analysis explained variance is higher in comparison with Library-based searches. We add comments in the Methods part, highlighting this issue.

- Given that the protein amount of a single mammalian cell (like HeLa cell) is as low as 150 pg, and accounting for inevitable losses in sample preparation including protein digestion, the authors could utilize 100 pg standard sample to evaluate the workflow method to enable true single-cell proteomics.

We thank the reviewer for this important question. Indeed, the protein content reported for mammalian cells varies a lot and depends on cell-type, cell state and size. To explain ourselves, we decided to focus our methodological comparisons on 250 pg as this reflects an ideal case scenario for a HeLa cell and gives a reference what could be the maximum number reachable from a true single cell. Aiming to answer the reviewers question and to give our readers an overview of proteome coverage to expect using the settings described in our study we added an additional dataset with injections ranging from 10 ng (corresponding to our library size) down to 50 pg corresponding to a smaller cell (Point-to-point figure 10). Our data shows still excellent coverage of 5752 protein groups on average from 150 pg. In line with the reviewer's suggestion, this is a bit more but very close to our result from true single cells (see updated single cell section in the main manuscript). We expect slightly higher numbers from the bulk digest since a) it's fully reduced and alkylated while single cell sample are not b) there is no trypsin contamination or potential sample losses.

Point-to-point figure 10: 50 pg – 10ng of HeLa peptides from the very same diluted bulk digest at 5ng/μL were injected each. Peptides were separated at 50 SPD throughput and data was recorded in DIA mode on the Orbitrap Astral MS and analyzed in direct DIA+ mode with or without usage of the method evaluation option as indicated using Spectronaut 18. Circles indicate identified protein groups (blue) or precursors (pink) at 1% FDR in technical replicates, bars indicate their means, while error bars indicate standard deviations with n=3.

Action taken: We included an additional dataset with HeLa dilution injections down to 50 pg and added Point-to-point figure 10 as Supplemental Figure 14. We discussed it in the first results section of the revised manuscript.

- The authors have overstated the protein identification results and should temper their claims, especially in the Abstract, Introduction and Discussion sections. Protein identification levels appear to fluctuate due to the use of carriers, cell size selection, and different gradients.

We agree with the reviewer that the given number in the title and abstract reflects the maximal numbers reached. We revised all sections to take care that they clearly state in which case the given numbers were reached. Furthermore, we added the full range of protein groups found depending on the cell-size and search strategy to the abstract to avoid any misleading phrasing. While only 50 SPD gradients were used for real single cells, we believe that the differences in ID numbers seen based on used tailored libraries and gradient length are part of the main message of our study on 250pg stocks clearly depicted in Figure 1 and 2 of the revised manuscript. Based on these results we chose the preferred LC-MS settings for our single cell measurements.

Of note, we further added an additional single cell dataset to the revised manuscript (Figure 6) reproducing the average proteome coverage initially reported and showing the strong dependency on the used cell-line. maximal number of 5300 protein groups for 2 different cell lines and even when using a trap-column.

Action taken: An additional single cell dataset was added to the revised manuscript. All sections mentioning protein numbers obtained from single cells were revised to avoid confusion between obtained maximal numbers and average numbers.

Reviewer #1:

Remarks to the Author:

The authors have addressed most of my questions. I only have minor comments left:

- The FDR formula used introduces a bias. Here I would like to refer to the recent excellent work by Bo Wen, Uri Keich and colleagues <https://doi.org/10.1101/2024.06.01.596967>, which details on why this formula should not be used. I would suggest to replace the FDR formula with the one I suggested in my review or with formula (1) in the above preprint. The true FDR in the benchmarks shown in the present manuscript is ~1.6-1.8%, if a correct formula is used, which is OK here.

Thank you for carefully going through our revised manuscript and your positive & constructive feedback.

We replaced the formula with the one from the above-mentioned literature as suggested. The obtained results using a shuffled DB are indeed exactly in the range that was expected by the reviewer. We adopted Figure 3 accordingly as shown below:

Point-to-point figure 1: Adopted Figure 3 from the main manuscript with FDR calculated as suggested by Wen et al., bioRxiv 2024.

- I don't think lower numbers in directDIA+ mode (page 4:128) indicate more stringent FDR control or improved data quality, but they could rather reflect the presence of global FDR control on top of run-specific one – I can only speculate about this, I am not familiar with the specifications of Spectronaut settings. But this alone cannot on itself be used as evidence about FDR or quality.

We agree with the reviewer that the global FDR control on top of the run specific one is likely the reason behind the slightly lower numbers with matching allowed. We further agree this is no evidence for FDR quality and rephrased the according passage accordingly as follows:

“Of note, we identified a total number of 6,126 protein groups from HeLa using directDIA+ in method evaluation mode and only 6,017 using directDIA+ with matching replicates. We hypothesize this is reasoned by a global FDR control applied over all 3 replicates on top of the run specific one while only very few additional proteins could be matched in those technical replicates: Of those 6,017 total proteins, 5,989 were found in all replicates, indicating an excellent data completeness (Supplemental Figure 1 A).”

To assess the influence of Spectronauts FDR parameters on ID numbers, we decided to apply an additional q-value filter to all the benchmark studies aiming for more realistic and conservative numbers:

We applied a cutoff of 0.01 for the run-wise Protein QValue and we additionally checked the results after applying stringent filters as suggested by Baker et al (<https://pubs.acs.org/doi/10.1021/acs.jproteome.3c00671>, there for previous Spectronaut versions). There they basically set all cutoffs to 0.01.

Please see the results obtained for 250 pg HeLa (Point-to-point figure 2), comparing default settings to the additional protein Q value cutoff, numbers do not change in direct DIA mode and drop only slightly for the library search. Using the most stringent settings, numbers drop by ~3 % for runs in DirectDIA with matching, and by ~16% for library searches. Very similar relative drops in ID numbers were observed for single cell runs. We updated the relevant figures accordingly in the revised manuscript and added the detailed search parameters to the methods part.

Point-to-point figure 2: 250 pg HeLa runs at 50 SPD as shown in the main manuscript, analyzed with default settings or with more stringent settings in Spectronaut 18.6, as indicated.

- Just to confirm, Supplementary Figure 15 is generated using non-normalised intensities? Please also indicate what sample & amount is being measured.

The data from Supplemental Figure 15 was generated from normalized intensities. To see if there is a general shift in intensity reasoned by FAIMS usage we additionally ad the non-normalized intensities to the supplemental figure and as shown below:

Point-to-point figure 3: Adopted Supplemental Figure 15 with a second panel showing non-normalized data.

- Supplementary Figure 19: median 75% CV is huge even for single cells. Please check that this refers to normalised data and maybe check intensity histograms for ubiquitous proteins in R to verify that Spectronaut normalised things correctly here (no need to include this in the manuscript though, if all checks out).

The shown data refers to normalized data. However, we found a mistake: CV were calculated using 66 cells and blanks. We corrected this mistake here and made sure that this didn't happen to the other datasets. We remove blanks for CV calculation, and got CV of 65%, which is still much higher in comparison with, for example, TE and hPSC datasets, where the median CV is 44%. We hypothesize that the CV is huge in that figure as it shows combined data from cells of different size fractions. When separating them into the size fractions used in the study the CV lowers to 52% (Point-to-point figure 5) which is in the range we expect for single cells.

Point-to-point figure 4: CV distribution for all 66 cells

Point-to-point figure 5: CV distribution of cell divided according to their size as indicated from 15-20, 20-25 or 25-30 μm .

Reviewer #2:

Remarks to the Author:

After carefully review the revised manuscript, I think the authors have addressed all the reviewers' concerns and make a good improvement to the manuscript. Therefore I agree that this manuscript should also be published on the Nature Method.

We thank the reviewer for carefully going through our revised manuscript and the positive feedback.